


# Modeling the integrated framework of complex water resources system considering economic development, ecological protection, and food production: A practical tool for water management

Yaogeng Tan[1,2], Zengchuan Dong[1], Sandra M. Guzman[2], Xinkui Wang[1], Wei Yan[3]

1. College of Hydrology and Water Resources, Hohai University, Nanjing 210098, China
2. Department of Agricultural and Biological Engineering, Indian River Research and Education Center, University of Florida, Fort Pierce, FL 34945, United States.
3. School of Geographic Sciences, Xinyang Normal University, Xinyang, 464000, China
Correspondence to Zengchuan Dong (zcdong@hhu.edu.cn)

**Abstract:** The rapid increase of population and urbanization is accelerating the consumption of the water resources that play an essential role in economic development, ecological protection, and food productivity (EEF). This study developed an integrated modeling framework to better identify the dynamic interaction, coevolution process, and feedback loops of the nexus across EEF systems by incorporating the multi-objective optimization and system dynamic (SD) models. The multi-objective model optimizes the water allocation decisions considering the adaptive status of both the whole system and each agent, while the SD model discloses the dynamics of the coevolution process and reciprocal feedback of the EEF system. The framework is applied to the Upper Reaches of Guijiang River Basin, China, in the context of interconnected systems considering the agents of economic development, ecological protection, and food productivity. Results show that the proposed framework enables the optimal water allocation decisions in each time step, and the SD model can adequately reveal the coevolution process and reciprocal feedback that differs in different stages in integrated, dynamic ways. The rapid economic growth intensifies the ecological awareness that cannot support such rapid development because of the conflict between environment and economic water uses. Once the economic growth rate decreased, water resources are able to support economic development because the ecological awareness is alleviated in this respect. The different water usages demonstrate the competitive mechanism, and the river ecological agent is the critical factor that affects the robustness of the model. The equal consideration of each water usage is the most beneficial to sustainable development. These results highlight the importance of water resources management considering multiple stakeholders and tradeoffs and give an insight into future dynamic changes of complex water systems.

## 1. Introduction

The rapid increase of the global population, urbanization, and economic development is accelerating the consumption of various natural resources (Zhang et al., 2018; Luo and Zuo, 2019). As one of the most important natural resources, water resources play an indispensable role in socio-economic development and food productivity, which is the fundamental condition of people's lives (Walter et al., 2012; Yang et al., 2019). Recently, the accelerated consumption of water resources impacted by human beings makes it vulnerable to ecological protection (Bei et al., 2009; Yang et al., 2019; Tan et al., 2019). In many regions, social activities have led to an enormous demand for water resources, which may negatively influence future population and development because it is at the expense of the environmental damages. There has been an increased interest in terms of "sustainable development" in respect of water resources whose new target aims to achieve effective and balanced uses of water resources that serves for





industrial, agricultural, and ecological practices and promote inclusive economic development (United Nations, 2014). Thus, it is more important than ever to detect the sustainable balance across the water needs of human for
economic development and food production, as well as those of the environmental protection (Baron et al., 2002; Falkenmark, 2003; Rockstrom et al., 2009; Yaeger et al., 2014; Perrone and Hornberger., 2016). The accelerated consumption of natural resources and the complex interactions across those needs challenge the achievement of this goal. Thus, the need to study the new approaches of the integrated framework motivates the current research on how the water resources system performs more properly by incorporating the interaction and coevolution processes of the
connected networks (Yaeger et al., 2014; Thompson et al., 2013; Wagener et al., 2010).

Water resources are the essential components of social development, food security, and environmental protection that are the core content of people's lives (Hunt et al., 2018; Uen et al., 2018; Perrone and Hornberger., 2016; Feng et al., 2019). Recently, the significance of water resources has been highlighted in the context of interconnected systems intensified by human beings considering those communities (Feng et al., 2019; Perrone and Hornberger.,
2016; Sivapalan et al., 2012). The goal of this context was to discuss how to achieve a systematic approach of both economic development and food security, based on environmental protection (Brundtland, 1987; Feng et al., 2016). The core content of sustainable development of water resources is to serve both socio-economic and ecological components (Flint, 2004; Konar et al., 2016; Rogers et al., 2002). To illustrate, socioeconomics also has underlying interconnections with industrial and agricultural practices. These practices require water resources to make profits.
Agriculture is the largest consumer of freshwater and contributes to food and crop production, and there is no doubt that food is the most fundamental condition of human survival (Li et al., 2019). Moreover, food production is also the economic income sources for farmers. Likewise, environmental stewardship is a complex system that maintains essential function and biodiversity in freshwater sources, vegetation, and ecological processes. Thus, the utilization of water in complex networks has an impact on the corresponding interconnected processes. In this view, the "nexus"
term emerges to reveal the framework of interlinked systems.

The "nexus thinking" was first conceived by the World Economic Forum (2011) to promote and discuss the indivisible relationships between the multiple use of resources, providing the universal rights of water, energy, and food (Hoff, 2011; Biggs et al., 2015). Furthermore, the framework of water-energy-food (WEF) nexus is propelled, which has drawn extensive attention (Allam and Eltahir, 2019; Sarkodie and Owusu., 2020). WEF refers to the
complex interlinkages among these three items to pursue sustainable development (Mabhaudhi et al., 2019). "Nexus thinking" is essentially the coupling of interconnecting systems that have been investigated in numerical frameworks, especially in integrated water resources management and socio-hydrology, presenting a new way for water management based on multi-stakeholders (Eum et al., 2012; Yarger et al., 2014). Apart from WEF nexus, other aspects of "nexus thinking" was also conceived by researchers and also contribute to sustainable development, such
as land use–water–energy linkages, of which the food is a core component (Howells et al., 2013; Ringler et al., 2013). Hellegers et al. (2008) outlined the concept of sustainability by combing water, energy, food, and environment, which can be regarded as the nexus thinking of water-energy-food-environment. This literature presented the urgency and necessity to assess four components' interaction to minimize the negative effects and enhance the synergistic benefits. Shahzad et al. (2017) stressed that water and energy are closely interlinked, and utilization of both resources will lead
to an increase in $CO_2$ emissions and environmental risks, and fulfill future sustainability by energy-water-environment nexus. Feng et al. (2016; 2019) outlined the framework of water, power, and environment systems (WPE nexus), disclosed their coevolution and response linkages of these three items, and gave a vital reference for policymakers.



The abovementioned literature has provided insights into the impacts of human society on hydrology, or, conversely, the hydrological cycle on socio-economy. However, the core content of nexus thinking based on interconnected systems is still the dynamics of their interaction and its quantification (Yaeger et al., 2014; Wagener et al., 2010; Collins et al., 2011). For example, Liu et al. (2007a) stressed that for water resources management in a watershed scale, the human and natural systems should be considered as integrity to deeper understand the coevolution and interaction process. The internal responses of a specific nexus system driven by external changes usually manifest in complex and nonlinear ways (Liu et al., 2007b). Thus, the water resources systems, coupled with human and natural systems, should be treated in a dynamic and integrated way to identify its coevolution and feedback process (Sivapalan et al., 2012). The complex and integrated water resources systems usually contain multiple agents (or water users), and an integrated and dynamic modeling approach is, therefore, helpful for tradeoff assessment across multi-objective of those agents, and further allows for the decision-making processes. Although the previous literature presents the way to understand and represent the decision-making analysis within the coupled dynamic systems, the integrated dynamic modeling still needs further investigation in terms of dynamic coevolution processes and feedbacks (Yaeger et al., 2014). As noted above, ecological protection, economic development, and food security are the key factors that consume water resources and are an essential part of human lives and sustainable development. Thus, this need motivates the current study that develops an integrated and dynamic framework considering those factors as a coevolving system.

The objectives of this paper are, therefore, (1) to develop an integrated modeling framework that couples the water uses across the economic development, ecological protection, and food production in a complex system perspective and explore the dynamics under external changes, (2) to apply the framework by introducing the multi-objective model and system dynamic model, in order to explore the optimal water allocation decisions, and their dynamic coevolution and feedback of the case study of Upper reaches of Guijiang River Basin, China, and (3) identify the model uncertainty to assess the various tradeoffs to stakeholders and recognize the main factor(s) that most influences the model robustness to improve the reliability of the integrated framework. In doing so, we are able to identify the dynamic coevolution and feedback of complex water resources systems for sustainable water resources management communities.

## 2. Outlines of the integrated modeling framework

Recent studies of water resources systems are gradually in the trend of systematic and integrated approaches to better understand the tradeoffs between water resources consuming sectors, such as that of maintaining people's lives, industrial and agricultural production, ecological protection, and hydropower, etc., (Moraes et al., 2010; Yaeger et al., 2014). Optimization models are one of the most indispensable approaches that solve the problems of the integrated model because the water utilization across multiple sectors is usually conflicted with each other. This study developed the systematic modeling approach of water resources across economy, ecology, and food (EEF) agent because these three items are the essential items of sustainable development, and they are also the primary consumer of water resources (Hunt et al., 2018; Uen et al., 2018; Perrone and Hornberger., 2016; Feng et al., 2019). The theoretical framework of EEF nexus is shown in **Fig.1**. An integrated modeling framework that considers systematics and optimizations, where economic agent is driven by external changes such as economic development level in different time steps, are also often acted as the alternative to give insight on how external changes drive the interactions and dynamics of the integrated water resources system (Secchi et al., 2011; Yaeger et al., 2014). The external changes of socio-economy can be addressed by the pendulum model outlined by Van et al. (2014) and Kandasamy et al. (2014).





Kandasamy et al. (2014) stressed that the term "pendulum swing" refers to the shift in the balance of water utilization
between economic development and environmental protection. It has the periodic changes that can be classified into
several stages in a long-term period. In short, it can be classified into the "initial" stage that productivity is about to
emerge, "developing" stage that production activities are negatively affecting the environment, and "environmental
protection" stage to which environmental issue is paid great attention. The detailed description of the "pendulum
model" can be found in **Supplementary material S1**.

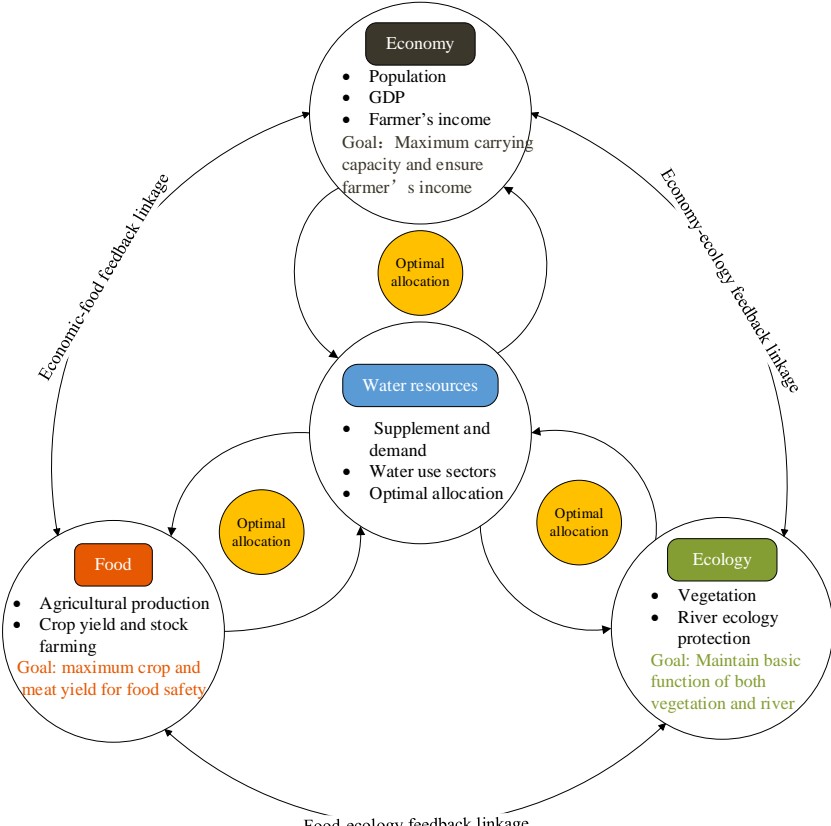


<p style="text-align:center">**Fig.1** Theoretical framework of EEF nexus</p>

The integrated modeling framework of water resources system comprises two models: the system multi-
objective optimization model that generates the water allocation scheme that considers the needs of each agent, and
system dynamic model that discloses the dynamic coevolution process and interactive feedbacks of the whole system.
The overall research framework of the integrated system considering two models is shown in **Fig.2**, and the detailed
model description is provided in Section 3. The external drivers of the whole nexus system are the changes of the
development level of socio-economy that can be outlined by pendulum dynamics (Kandasamy et al., 2014). The
system dynamics, coevolution process, and feedbacks affected by the external drivers can be assessed through both
coevolution trajectories and impact-response trajectories (i.e., feedback loops) in terms of water supply and demand
of each agent, carrying capacity (economy agent), farmer's profit (economy agent), crop yield and meat production





(food agent), vegetation water (ecological agent), and streamflow water (ecological agent). Each agent comprises multiple issues (as shown in **Fig.2**), and the detailed description and equation can be seen in **Supplementary material S2**.

From the perspective of systematics, the optimization process of both agents and the whole system can be
explained by complex adaptive system (CAS) theory (Holland et al., 1992). The term "self-adaptive adjustment" is derived from CAS, where the corresponding agents can attain the status of mutual coordination and achieve their own goals by improving their behaviors. CAS stresses that each agent has the ability of both self-studies to improve their behaviors and the response mechanism to the external changes, becoming the more potent agents to adapt to the external changes. According to Holland et al. (1992), the optimization of each agent is namely the corresponding
self-adaptive process that tends to be more potent. The process of self-adaptive adjustment can attain the best status of each agent, and the best status of the global system will then achieve. The optimal model is, therefore, to accomplish the process of adaptive status of both each agent and overall system.

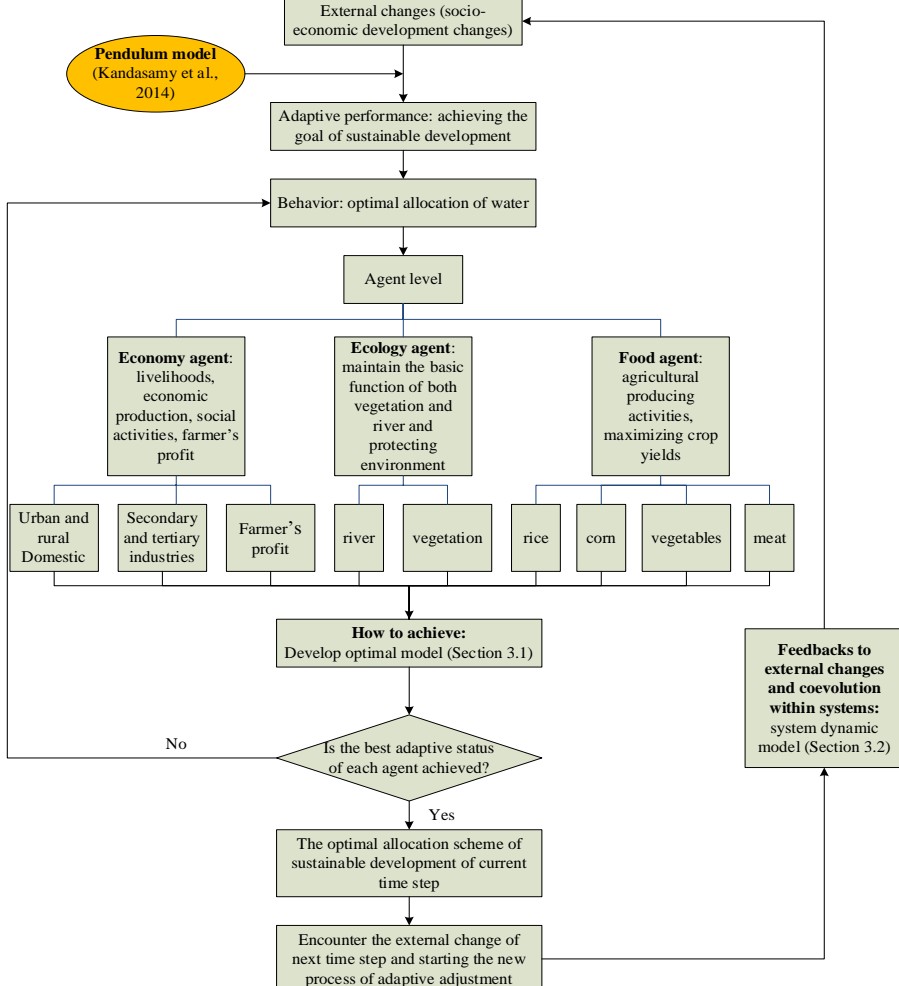

**Fig.2**  Overall research framework of the integrated modeling approach



150       The two models are run interactively to generate the trajectories of coevolution and feedbacks that reveal the dynamics with the output of the optimization model. The optimization model provides input decisions of the system dynamic model. As shown in **Fig.2**, The external changes in a certain time step trigger the adaptive adjustment of each agent that is reflected by the optimal allocation process of water resources across multiple water users. In other words, optimal allocation process can help each agent achieve its best status under changing external conditions.

Then, the system dynamic model that reveals the mutual interactive relationship can impose the feedbacks and calculate the assessment parameters (e.g., carrying capacity, food yield, etc.) to each agent through optimal water allocation schemes from the external changes of the current time step. Then, the assessment results can also be incorporated into the decision-making process in the next time step of external changes. The optimal decision of the current time step can act as the beginning of the new external drivers of the next time step. In other words, both the

optimization and system dynamic process will be reproduced to generate the feedback and optimal decision of the next time step. Therefore, the dynamic coevolution trajectories of the water resources system will be generated by connecting the assessment results of corresponding parameters to each agent of each time step.

## 3. Modeling framework application

### 3.1 Optimization approach of the integrated system

#### 3.1.1 Model conceptualization

      The framework of sustainable development theory presented in **Fig.1** is of great significance by applying it in a specific region or watershed. For example, in a water system inside a watershed or a region, there are multiple water supply projects that supply water to different water users. This system in a watershed is called a "large water resources system" (**Fig.3**a). It is subdivided into multiple sub-watershed or subregions that are called "subsystems" (**Fig.3**b).

In this case, reservoirs can provide not only socio-economic developments but also environmental impacts. They are constructed across the rivers to supply water for the whole region or watershed but are also most likely to cause negative impacts on the natural streamflow of rivers, which will deteriorate the instream ecological environment (Yin et al., 2010; 2011; Yu et al., 2017). Therefore, reservoirs should be considered individually to target the river ecology concerns.

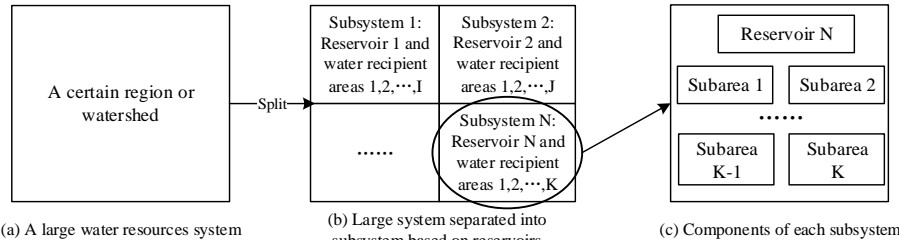

         (a) A large water resources system      (b) Large system separated into subsystem based on reservoirs      (c) Components of each subsystem

175       **Fig.3**    Water resources system and its decomposition

      To fully consider the river ecological health, the whole system is separated into subsystems that contain one individual reservoir and its several corresponding water recipient areas (**Fig.3**b) as there is usually more than one reservoir in a certain region. We call these subsystems as "reservoir supply subsystem". A subsystem can be further

separated into the smallest unit: a reservoir and each water recipient region (or called "subarea") (**Fig.3**c). In this view, the total system of the water resources in a certain region (watershed) can be divided into several subsystems or subareas that consist of a three-level hierarchical structure. According to the theoretical framework of sustainable





development of EEF presented in Section 2, each agent has its own goals, and they can be distributed to each subarea
(with the objective of food, socio-economy, and vegetation) and reservoir (river ecology) (**Fig.4**). Therefore, we can
coordinate these objectives to achieve sustainable development by setting up multi-objective optimal model.

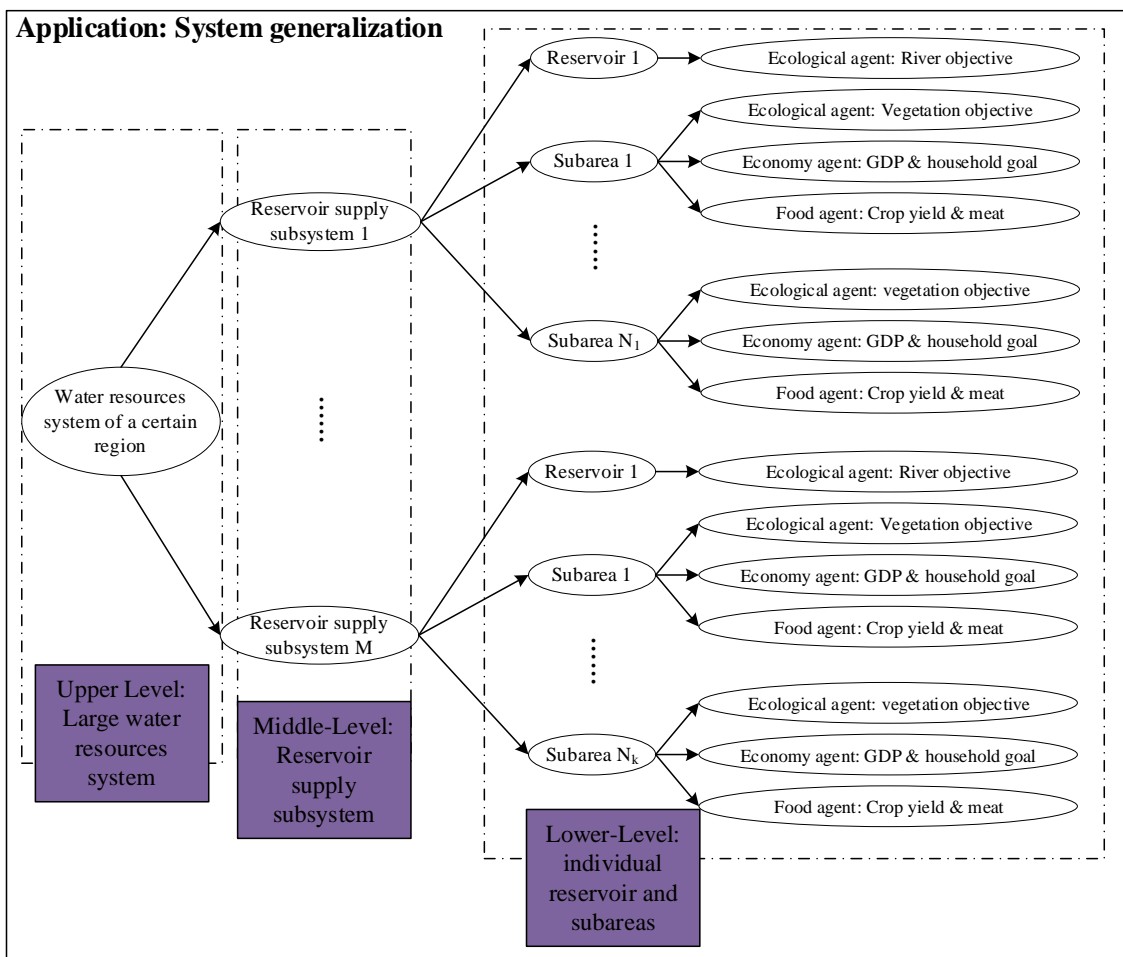

**Fig.4**  Application of sustainable development theory on watershed or region

*3.1.2 Objective function*

(1) Economic agent

The objective function is presented based on each component of the EEF nexus. The goal of the economic agent
is aiming at increasing revenue of secondary and tertiary industries, as well as maintaining human wellbeing. It can
be reflected by the minimum household and industrial shortage and is expressed by the following normalized
nonlinear equation:

$$F_{ecnmy} = \frac{1}{T} \min \sum_{k=1}^{K} \sum_{t=1}^{T} \left( \frac{WD_{ecnmy,kt} - WS_{ecnmy,kt}}{WD_{kt}} \right)^2 \qquad (1)$$

where $F_{ecnmy}$ is the objective function of the economic agent. WD and WS are total water demand and supply





(including reservoir and other water projects) of this agent. T is the total number of the time step. Subscript k and t are the number of subarea and time step, respectively. Water demand of this agent includes household and industrial water demand, by multiplying the water use quota with economic index (population for household, GDP for industry, see **Supplementary material S2**). It should be noted that farmer's income affiliated in the economic agent is greatly
related to food production. Thus, this goal will be discussed in food agent.

(2) Ecological agent

Economic development should not be excessive because it may be at the expense of the damaging ecological environment, which is inconsistent with the concept of sustainable development. River ecology should be especially paid attention to because reservoir construction is alternating the natural flow and will deteriorate the river's function
irreversibly, further affecting the ecological environment negatively (Poff et al., 1997). The artificial intervention in the natural flow regime is a crucial factor of the severe deterioration of river ecosystems (Shiau et al., 2013; Tan et al., 2019). Vegetation, similar to the river environment, is also an indispensable part of ecology because it produces oxygen to improve air pollutions and purifies water bodies. Therefore, water resources support is also essential for maintaining the physiological function of vegetation and river health. The objective of the ecological agent is
reflected by maintaining both aspects.

$$F_{eclgy} = \frac{F_{veg} + F_{riv}}{2} \tag{2a}$$

where $F_{eclgy}$ is the objective function of the ecological agent, and

$$F_{veg} = \frac{1}{T} \min \sum_{k=1}^{K} \sum_{t=1}^{T} \left( \frac{WD_{veg,kt} - WS_{veg,kt}}{WD_{veg,kt}} \right)^2 \tag{2b}$$

$$F_{riv} = \min \frac{AAPFD}{5} \tag{2c}$$

where the subscript "veg" and "riv" represents vegetation and river ecology. AAPFD is the abbreviation of "amended annual proportional flow deviation" that reflects the river's health degree to assess the diversity of fish species (Gehrke et al., 1995; Ladson and White, 1999). This parameter is defined to reflect the difference between observed (natural) and actual streamflow. According to Ladson and White (1999), the value of AAPFD ranges from zero (very healthy) to five (entirely damaged). Details can be seen in **Supplementary material S2**. Here, we divided it by five
to normalize AAPFD and make it range from zero to one. Meanwhile, $F_{eclgy}$ is also normalized by getting the average of $F_{veg}$ and $F_{riv}$.

(3) Food agent

The goal of the food agent is to maximize food production and is the indispensable condition of increase primary industry products and farmer's income. Also, food is one of the most fundamental prerequisites for people's survival.
The mathematical expression is presented as follow:

$$F_{food} = F_{crop} + F_{livestock} \tag{3a}$$

where



$$F_{crop} = \max \sum_{n=1}^{N} \left( \frac{Y_a}{Y_p} \right)_n \tag{3b}$$

$$F_{livestock} = \max \sum_{l=1}^{L} Y_L \tag{3c}$$

where N and L are the total number of crops and livestock, respectively. $Y_a$ and $Y_p$ are the crop yield under the actual and ideal conditions, respectively. $Y_L$ is the total meat production. The farmer's income is presented as follow:

$$I = \frac{1000 \sum_{i=1}^{n} Y_i p_i}{N_r} \tag{4}$$

where I is the farmer's average income, $Y_i$ is the ith food (including crop yield and meat production). Food production is a significant component of both primary industry value and can measure farmer's income, because farmers sell

these foods to customers and get profits. We can see that the farmer's income is in proportion to food production. In other words, maximum food production is the prerequisite of the maximum income. The calculation of food yield is based on the Food and Agricultural Organization report No. 56 (FAO-56) (Allen et al., 1998). For details, see **Supplementary material S2**.

*3.1.3 Tradeoffs between objectives*

According to the crop yield equation based on FAO-56 (see Eq.(6) in **Supplementary material S2**), crop production that determines farmer's profit is directly related to irrigation water (FAO, 2012; Liu et al., 2002; Lyu et al., 2020), and the production of livestock is also in proportion to its water usage (see Eq.(7) in **Supplementary material S2**). Therefore, the maximum supply of crop and livestock water demand is the most critical condition for maximum crop yield or meat production. Thus, the normalized objective of food agent can be rewritten as:

$$F_{food} = \frac{1}{T} \min \sum_{k=1}^{K} \sum_{t=1}^{T} \left( \frac{WD_{food,kt} - WS_{food,kt}}{WD_{food,kt}} \right)^2 \tag{5}$$

where $WS_{food}$ and $WD_{food}$ are the irrigation or livestock water supply & demand. Similarly, the maximum satisfaction of industrial and household water demand can get the maximum profit and revenue as well as human wellbeing, which is the same as the minimum water shortage. The same also applies to vegetation water.

As can be seen in objective functions, three benefits are set minimum (Eqs.(1)(2a)(5)), which may contribute to

the conflict between objectives, especially ecology and economy. The tradeoffs across EEF nexus can be reflected by Pareto frontier that can describe a set of non-dominated optimal solutions that any one of these three objectives are unable to be improved unless sacrificing other objectives (Reddy and Kumar, 2007; Feng et al., 2019; Beh et al., 2015; Burke and Kendall., 2014). We can reclassify all the water users from each of the three agents into two categories: Instream and off-stream water users (Hong et al., 2016). River ecological water demand can be regarded

as an instream water user, and all others can be considered as off-stream water users. Therefore, according to the objective function expressed by Eqs.(1),(2), and (5), the weighted objective function can be rewritten by:





$$\min F = F_{ecnmy} + F_{eclgy} + F_{food} = \alpha \left( F_{ecnmy} + F_{veg} + F_{food} \right) + \theta F_{riv}$$

$$= \sum_{j=1}^{J} \sum_{k=1}^{K} \sum_{t=1}^{T} \alpha_j \left( \frac{WD_{jkt} - WS_{jkt}}{WD_{jkt}} \right)^2 + \theta \frac{AAPFD}{5} \tag{6}$$

where ($F_{ecmny}$+$F_{veg}$+$F_{food}$) is off-stream water users, and $F_{riv}$ is the instream water users. The subscript j is the index of the off-stream water users, respectively. j=1,2,3 represents socio-economic, food, and vegetation water usage,

which corresponds to the subscript "ecnmy", "eclgy" and "food". α and θ are weight factors and $\sum_{j=1}^{J} \alpha_j + \theta = 1$.

Previous literature demonstrated the optimal solution shaped like Eq.(6) is Pareto-optimal because of the positive weights and concave objectives, and the non-dominated sorting process is used to find the optimal solution of Eq.(6) because the characteristic of either concave or convex is difficult to be proven (Marler and Arora., 2009; Feng et al., 2019; Goicoechea et al., 1982; Zadeh, 1963). For each given combination set of α and θ, the optimal solution can be

attained by decomposition and coordination (DC) principle and dynamic programming (DP) (see section 3.1.5).

The tradeoff across objectives is reflected in the values of multiple sets of weighting factors $r = \left( \alpha_1, \alpha_2, \alpha_3, \theta \right)^T$, revealing different decision-makers' preferences. Considering that the contradictions also occur in off-stream water users, the balanced priority should be addressed to consider each off-stream water users (Casadei et al., 2016), that is, $\alpha_1 = \alpha_2 = \alpha_3$. Therefore, the tradeoff and decision preference between instream and off-

stream is reflected by the different values of θ (0≤θ≤1). The larger value of θ represents more concerns about river ecology. In this study, the parameter θ is initially set as 0.5 to give an equal consideration of both instream and off-stream water usage. It should be noted that this weight combination is one possible set that considers the equal use of instream and off-stream water uses, and different weight of weighting factor reveals the preferences of stakeholders. Different vectors of $r$ can affect the performance of EEF nexus and are used to assess the uncertainty and robustness

of the model to improve its reliability (see Section 6.1 & 6.2).

*3.1.4 Constraints*

The model constraints include the connection of subsystems, the water balance equation, and the upper and lower limits. The details are found in **Supplementary material S3**.

*3.1.5 model solution*

The EEF model of water resources sustainability is a compound system that is classified into multiple hieratical structures (**Fig.4**). Therefore, the model solution of this structure should be solved by systematical analysis techniques, such as Dantzig-wolfe decomposition technique (Deeb and Shahidehpour, 1990), Generalized Bender Decomposition (Rabiee and Parniani., 2013), aggregation-decomposition (AD) (Tan et al., 2017) and decomposition-coordination (DC) (Li et al., 2015; Jia et al., 2015). Considering DC method can reduce the system dimension to save computing

time, and optimization order among each subsystem is arbitrary, this study uses DC method to solve this sophisticated model. The total procedure of both DC and DP is provided in **Supplementary materials S4**.

*3.2 Coevolution and responses of the integrated system based on system dynamic (SD) model*

*3.2.1 Coevolution mechanism of the integrated EEF system*

Water resources provide the resources support for agriculture (food agent), industry (economy agent), and


environment (ecology agent). These components can, respectively, provide the crops and meat to ensure food security, make profits, and make humans and nature co-exist harmoniously. The mutual relations among the three components of an EEF nexus determines the coevolution process (Feng et al., 2016). According to the framework of EEF nexus presented in section 2, the coevolution and responses of EEF nexus are shown in **Fig.5**.

As shown in **Fig.5**, the socio-economic development, along with the population and GDP size, will undoubtedly
increase (Biggs et al., 2015; Duan et al., 2019), which will be reflected in an increase in water demand (I). The increased population and GDP is namely the external drivers of the integrated framework. However, the ecosystem will be damaged due to the volume of water that will supply those increased population needs (II). Therefore, the optimization model presented in this study can provide information to coordinate the nexus between systems, provide a water allocation scheme based on each agent's water requirements, and maintain the ecological health of rivers and
freshwater sources (III). The population and GDP growth rate cannot increase infinitely because regional water resources are usually unable to carry a continuously exponential growing population size and GDP. We call this term "carrying capacity" to describe the rate of socio-economic development under certain water resources conditions (Yang et al., 2019; Wu et al., 2018). It is determined by the amount of actual water supply and allocation in a certain year. The carrying capacity can reflect the development status and affect the predicted socio-economic indexes (IV).
It can give references for policymakers for comprehensive urban planning and can influence the process of coevolution and feedback of EEF nexus (V). In this study, we use the concept of "overload index" to illustrate the relationship between carrying capacity and predicted economic index (mainly for population and GDP) and is expressed as follow:

$$OI = \frac{PI}{CI} \tag{7}$$

where OI, PI, CI is the overload index, predicted economic indicator, and carrying economic indicator (i.e., carrying capacity). The overload index can be classified into five levels based on the value of OI and shown in **Table 1**. This feedback loop indicated that the rapid growth of the economy would deteriorate ecological health because the limited water resources in a certain area cannot afford the increasing socio-economy. Additionally, environmental health is an indispensable element of sustainable development. It will further decrease the carrying capacity, and the socio-
economy will be negatively influenced, stimulating the policymakers to readjust the scale of socio-economy.

**Table 1**  Classification of the overload index level

| Value of OI | Overload index level |
|:---:|:---:|
| ≤0.7 | Well-loaded |
| 0.7~1.0 | Rational-loaded |
| 1.0~1.3 | Minor overload |
| 1.3~1.5 | Moderate overload |
| >1.5 | Serious overload |





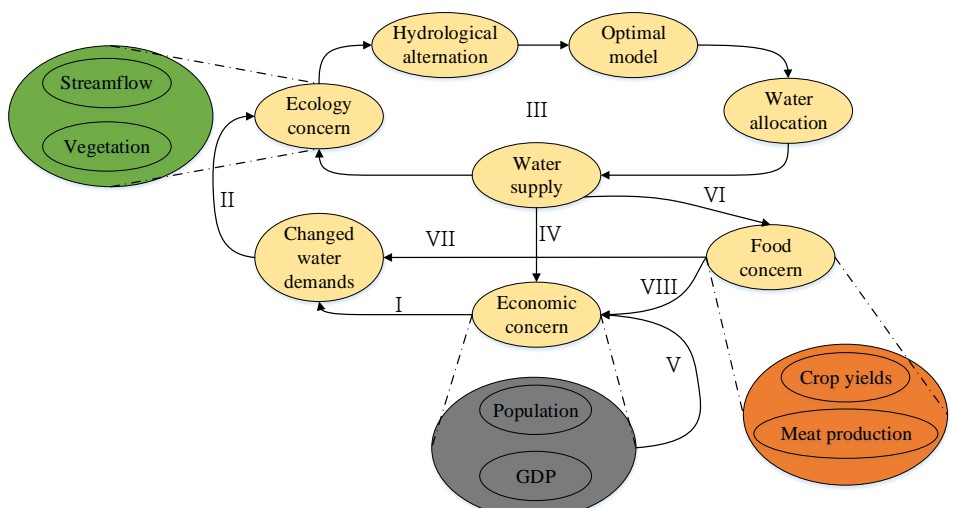

**Fig.5** Coevolution and feedback mechanism of EEF nexus

The ecology-food nexus reflects another feedback of the EEF nexus. Agricultural water is the largest water
consumer and is deeply affected by rainfall and potential evapotranspiration. According to Allen et al. (1998), more
evapotranspiration will cause more agricultural water demand (VII), and water supply pressure from water projects
will increase subsequently. However, if the rainfall increases, there will be less water supply pressure. Otherwise, the
increased water supply from reservoirs will alternate the natural flow, which will deteriorate the river's ecological
health and drive the optimal model to adjust the water allocation scheme (VII-II-III). Afterward, the agricultural water
supply will affect food production (VI), which is similar to the effect that the economy-ecology nexus reflects.
However, the socio-economical changes would also indirectly affect the food system and more than just rainfall and
evapotranspiration, i.e., the changes of economic concern will also drive the optimal model described in this study
and further influences the food production (I-II-III-VI). Besides crop production, stock farming is another source of
food for meat production and is also affected by this economy-ecology linkage. It should be noted that as food system
is the indispensable component for human lives and primary industry, the food production will directly affect the
changes of carrying population and farmer's income (VIII) and subsequently affects the feedback loop of economy-
ecology (I-II-III), further starting the new loop of whole EEF nexus.

### 3.2.2 Development of system dynamic (SD) model

From **Fig.5**, we can see that each component EEF nexus interacts mutually and are reciprocal causation. They
are interconnected by the changes in the water supply and demand system. To reflect the complicated and detailed
relationships and feedbacks based on **Fig.5**, system dynamic model (SD) (Forrester and Warfield., 1971) is presented
in this study. It is a well-established system simulation method for visualizing, understanding, and analyzing
complicated dynamic feedback systems that exhibit nonlinear, multi-feedback, and time-varying properties (Yang et
al., 2019).

According to the coevolution and feedback mechanism of EEF nexus shown by **Fig.5**, system dynamic (SD)
model is used to reflect the coevolution and feedback process by the cause-and-effect feedback loop that is the
inherent function of SD. **Fig.2** illustrated the EEF dynamics under external changes that affect its adaptive behaviors,





and this cause-and-effect feedback loop can embody the dynamics graphically. The feedback loop based on SD model is shown in Fig.6. The arrow represents the linkage between variables called feedback regulation in SD. Feedback regulation can be categorized into two types: positive and negative. Positive feedback regulation indicates that the changes in the certain factor will shift to the side of the changing trend of the related factor that would promote that change. In other words, a(n) increased/decreased independent variable results in the increased/decreased dependent variables. Positive feedback regulation may usually lead to the polarization of a whole system. Negative feedback regulation refers that the changing trend of a certain factor would reduce that change of the related factor(s), and the stable status of a whole system will eventually be attained. In SD model, the "+" and "-" symbols represent the positive/negative feedback and are marked beside the corresponding arrows. Positive and negative feedback regulations connect these variables, and several closed feedback loops are eventually formed. The positive/negative feedback loop is marked as the "+" symbol with a clockwise arrow and "-" symbol with a counterclockwise arrow.

For example, if the ecological issue is not considered, the increased population intensifies the household water demand, which further strengthens the water supply. Then the increased water supply will support more population, and it forms the positive feedback loop. However, the total integrated model comprises three adaptive agents (See **Fig.2**) in which the optimal approach achieves the best status. The Pareto frontier of the optimal approach reveals the conflict between ecological streamflow and social water use. Therefore, the increased water supply (ecological issue not consider yet this moment) intensifies the ecological awareness, resulting in the increased streamflow and further reduces actual water supply. The decreased water supply decreases the carrying capacity and further increases the overload index, and the final population size will be readjusted. This is a negative feedback loop. We can see that if the water supply increases at the same rate as water demand caused by increased socio-economic index, this feedback will be the positive feedback regulation that results in the polarization because the ecological water will be occupied and environmental protection will not be guaranteed. Therefore, the optimal model is used to attain a sustainable development goal, and it is used to generate a negative feedback loop to maintain a stable EEF system.

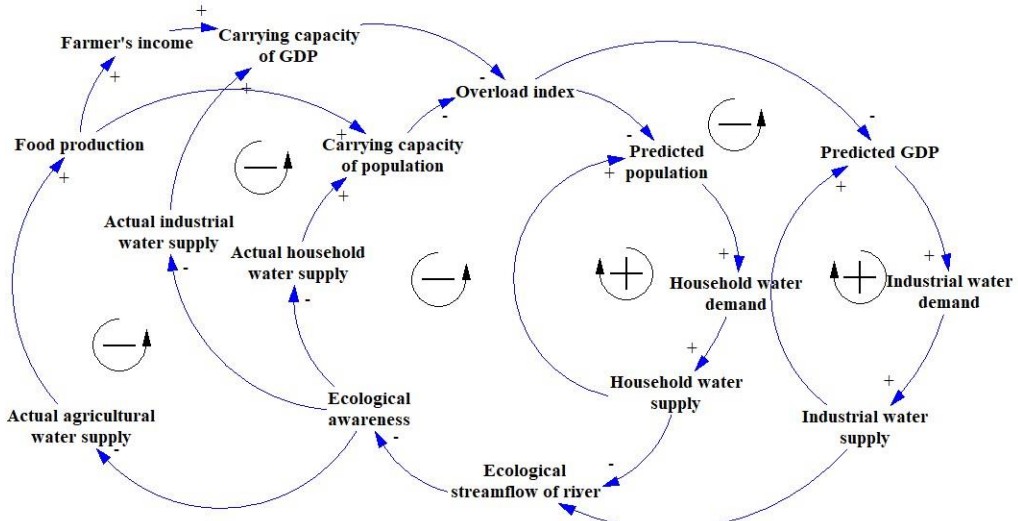

Fig.6 Cause-and-effect feedback loop of SD model


*3.2.3 Coevolution dynamics and feedbacks based on the modeling framework*

As shown in **Fig.2**, the trajectories that reflect the coevolution dynamics and feedbacks of the integrated system are generated by combining the optimal model with SD model by connecting the corresponding parameters of different time steps. The coevolution and feedback trajectories include the parameters of different agents to the nexus system, including carrying capacity, food production, farmer's profit, streamflow water, and water supply & demand, etc. These variables can be calculated by SD model where the main function is to figure out the mathematical relationships between variables based on the positive/negative feedback. The mathematical relationship can be

reflected by the total flowchart that consists of variables and linked arrows. The completed flowchart and mathematical relationship can be found in **Supplementary material S5**. But here, we have presented the variables and their mathematics that is greatly related to the result section (Section 5) and listed in **Table 2**. It should also be noted that some variables in SD model are called the "lookup" function that reveals the changes of some variables over time. In this case, the water allocation (supply) over time is calculated by the optimal model over external drivers

of socio-economy. Thus, water supply is the main lookup function and can act as the input of the SD model. The trajectories of corresponding parameters related to different agents can be figured out by the following mathematics (**Table 2**), and others are listed in **Supplementary material S5**.

**Table 2**    Mathematics of coevolution dynamics

| Variables | Notations | Units | Mathematics | Remarks |
|---|---|---|---|---|
| Water supply (includes overall and each agent) | | $10^8 m^3$ | Lookup function solved by optimal model | Water supply include household, (territory) industrial, food, and vegetation |
| Streamflow water | | $10^8 m^3$ | Lookup function solved be optimal model | |
| AAPFD | | | $AAPFD = \dfrac{1}{n}\displaystyle\sum_{j=1}^{n}\sqrt{\sum_{m}^{12}\left(\dfrac{Q_{mj}-QN_{mj}}{QN_j}\right)^2}$ | Q, QN are actual and natural streamflow; subscript j, m are year and month number, n is total year number |
| Objective function of economy agent | $F_{ecnmy}$ | - | ((Water demand for economy agent-water supply for economy agent)/water demand for economy agent)$^2$, see Eq.(1) | Water demand of this agent is calculated by multiplying predicted population//GDP with water use quota |
| Objective function of food agent | $F_{food}$ | - | ((Water demand for food agent-water supply for food agent)/water demand for food agent)$^2$, see Eq.(5) | Crop water demand is calculated based on FAO-56, see **Supplementary material S2** |
| Objective function of ecology agent | $F_{eclgy}$ | - | Eq.(2a)~(2c) | |
| Carrying capacity: | | people | Household water supply × 1000/(water quota for household × day of a certain | The unit of water quota for household is L/people/d |



| population | | year) | |
|---|---|---|---|
| Carrying capacity: GDP | $10^8$ yuan | (Industrial water supply + tertiary water supply)/Water consumption per 10000RMB of non-agricultural industry + Total value of primary industry | |
| Crop yield | $10^4$t | Crop yield is the nonlinear function of crop water supply and demand, and details can be seen in **Supplementary material S2**. | |
| Meat production | $10^4$t | A × water use of livestock + B | A and B are coefficient of linear regression, where water use and meat production of livestock can be found in statistics of water resources bulletin and socio-economic over years (Li et al., 2019). |
| Total value of primary industry | $10^8$ yuan | Food production × food price per unit | Food production is the summary of crop yield and meat production |
| Farmer's annual income | $10^4$ yuan per capita | Total value of primary industry/rural population | Eq.(4) |
| Overload index | OI | - | Predicted economic index/ Carrying capacity | Eq.(7) |

### 3.3 Sustainable development degree (SDD) assessment

The EEF nexus is a complex system with all ecological, economic, and food systems, or agents as we called in this study, affecting water resources. A proper EEF balance provides resource support to achieve sustainable development. Therefore, the three agents should be considered to evaluate the sustainable development degree. We selected the indicators listed in **Table 3** based on the three agents and are used to evaluate the impact of sustainable development.

**Table 3** Sustainable development evaluation index system of three agents

| Agent | Indicators | Property |
|---|---|---|
| | Overload index of population | - |
| | Overload index of GDP | - |
| Economy | Per capita GDP (RMB/people) | + |
| | Water consumption per 10000RMB of GDP (m³/$10^4$RMB) | - |
| | Farmer's income (RMB/people) | + |





| Food (Agriculture) | Meet production (t) | + |
| | Crop yield (t) | + |
| Ecology | Effective irrigation area for vegetation (km$^2$) | + |
| | AAPFD | - |

The property (+, -) of indicators denotes positive and negative indicators, respectively. The positive/negative indicators mean they have positive (negative) impacts on the corresponding agent and were termed as a development/constraint index (Yang et al., 2019). Considering the ranges of indicators listed in **Table 3** are different, they should be normalized before evaluation. The positive and negative indicators normalization is shown by Eq.(8a) 395 and (8b).

$$y_{ij} = \frac{x_{ij} - \min\limits_{i=1}^{m} x_{ij}}{\max\limits_{i=1}^{m} x_{ij} - \min\limits_{i=1}^{m} x_{ij}} \tag{8a}$$

$$y_{ij} = \frac{\max\limits_{i=1}^{m} x_{ij} - x_{ij}}{\max\limits_{i=1}^{m} x_{ij} - \min\limits_{i=1}^{m} x_{ij}} \tag{8b}$$

where $x_{ij}$ and $y_{ij}$ is the original and normalized indicator j in sample i, and m is the total number of samples. The entropy weight method is then adopted to calculate SDD, which calculates the information entropy of indicators that 400 reflect their relative change degree on the whole system (Wang et al., 2019). The information entropy of indicator j in sample i is expressed by:

$$E_j = -\frac{1}{\ln m} \sum_{i=1}^{m} d_{ij} \ln d_{ij} \tag{9a}$$

$$d_{ij} = \frac{y_{ij}}{\sum\limits_{i=1}^{m} y_{ij}} \tag{9b}$$

Finally, the entropy weight of each indicator is expressed by:


$$\omega_j = \frac{1 - E_j}{\sum\limits_{j=1}^{n} \left(1 - E_j\right)} \tag{10}$$

where n is the total number of indicators in a certain agent.

The SDD is calculated based on the coupling coordination degree (Sun and Cui, 2018), reflecting the degree of coordination of various factors or subsystems. In this study, SDD is calculated based on the coordination of three agents (EEF) and expressed by:


$$SDD = \sqrt{C_1 C_2} \tag{11a}$$





$$C_1 = \left[ \frac{ECNMY(t) \cdot ECLGY(t) \cdot FOOD(t)}{(ECNMY(t) + ECLGY(t) + FOOD(t))^3} \right]^{\frac{1}{3}}$$

$$= \left[ \frac{\sum_{p=1}^{P} \omega_{pj} y_{pj} \cdot \sum_{q=1}^{Q} \omega_{qj} y_{qj} \cdot \sum_{r=1}^{R} \omega_{rj} y_{rj}}{\left( \sum_{p=1}^{P} \omega_{pj} y_{pj} + \sum_{q=1}^{Q} \omega_{qj} y_{qj} + \sum_{r=1}^{R} \omega_{rj} y_{rj} \right)^3} \right]^{\frac{1}{3}}$$

(11b)

$$C_2 = \frac{1}{3} \left( ECNMY(t) + ECLGY(t) + FOOD(t) \right)$$

$$= \frac{1}{3} \left( \sum_{p=1}^{P} \omega_{pj} y_{pj} + \sum_{q=1}^{Q} \omega_{qj} y_{qj} + \sum_{r=1}^{R} \omega_{rj} y_{rj} \right)$$

(11c)

where ECNMY(t), ECLGY(t), and FOOD(t) are the coordination degree of economy, ecology, and food agent, respectively. P, Q, R is the total indicator number in economy, ecology, and food agent.

## 4. Study area and data sources

*4.1 A brief description of the study area*

Guijiang River Basin (GRB) is one of the most imperative branch basins of the Xijiang River Basin (XRB) in South China. XRB belongs to the typical karst area and is the second-largest river basin in China in terms of total runoff and also the third largest river basin in terms of total area. The mainstream of XRB are Nanpan River, Hongshui River, and Xijiang River in the upper, middle, and lower stream. Yujiang, Liujiang, and Guijiang are the main branch river of XRB (see **Fig.7**). The upper reach of Guijiang River Basin (UGRB) (24°6' ~25°55'N, 110°~111°20'E) is selected as a case study as it represents the highly conflicts between socio-economic growth and ecological protection in karst areas. Furthermore, reservoirs are widely constructed in UGRB to supply water for socio-economy but are likely to deteriorate the river ecological health by alternating natural flow (Yin et al., 2010; 2011). UGRB is also a karst area with a total area of 13,131 km², with about three million people. Also, UGRB has a total crop planting area of about 2,400 km², a total vegetation area of about 3,700 km², and yearly average precipitation of about 1600mm. UGRB is located in Guilin City and refers to eight administrative regions (or counties). Seven reservoirs are constructed in UGRB to provide water resources support for maintaining the development of socio-economy. The detailed parameters of seven reservoirs and their three-level hieratical structure, including subareas, are found in **Supplementary material S6**. Guilin city is both a heavy industrial city and a national major tourist city, and the population and economic development will keep rapidly increasing in the near future. It will exacerbate the conflicts between social development, food safety, and environmental protection, especially for water use of river ecological environment, resulting in severe ecological deterioration of the lower Guijiang River basin and even lower XRB. Therefore, how to achieve coordination and sustainable development in UGRB between these aspects is becoming a challenging problem in the upcoming years and is necessary to be solved.

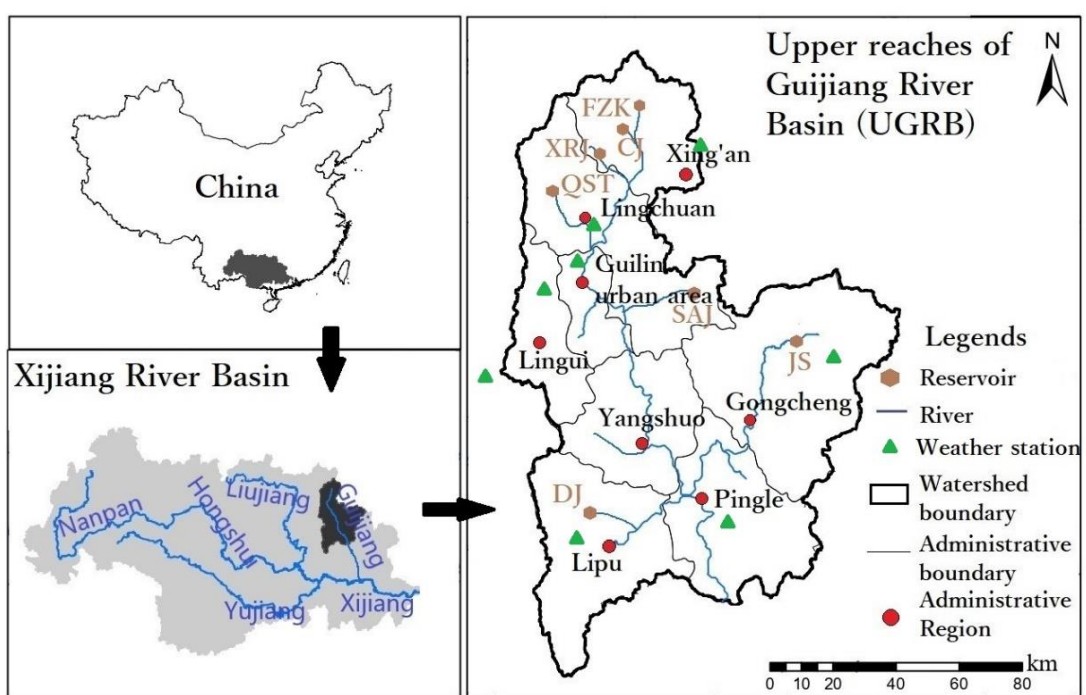

**Fig.7** A brief location of UGRB

*4.2 Datasets and parameter initialization*

Datasets of the case study include socio-economic, water use, land use, meteorological and hydrological data.
The major source of socio-economic data, including population and GDP, are the statistical yearbooks of both Guilin
City and Guangxi autonomous region from 2005-2014 (http://data.cnki.net). The Municipal Government of Guilin
City (MGGC) predicted population and GDP till 2040, along with per capita water use from the water industry
standard of the People's Republic of China, to predict the water demand of economic agent (Venkatesan et al., 2011).
These predicted economic indexes are exactly the external drivers of the whole integrated modeling framework (see
Section 2), and the corresponding growth rate in different stages are shown in **Table 4**. The sharply increased rate in
the second stage, which corresponds to the era that "heavy government policy support and investment" and
"population grow rapidly", is stressed in "pendulum model" by Kandasamy et al. (2014) (see Section 2 and
**Supplementary material S1**). The growth rate from 2031 to 2040 is lower compared with that from 2021 to 2030,
which corresponding to the era of "remediation and emergence of the environmental customer" as stated in
Kandasamy et al. (2014). Water use data include historical water usage and total water amount found in Guilin water
resources bulletin (2005~2014). Land use data contain the spatial distribution of crops and vegetations with a
resolution of 1km×1km that can be found in the Resource and Environment Data Cloud Platform, China Academy
of Sciences (REDCP-CAS) (http://www.resdc.cn). Meteorological data from 1956 to 2013, including daily average
wind speed, sunshine duration, maximum and minimum temperature, relative humidity, and precipitation, are found
in meteorological stations (http://data.cma.cn). The hydrological data from 1958 to 2013, including the monthly


inflow of each reservoir, can be found in hydrological stations. All the initialized parameters and the total index of the data sources can be found in **Supplementary material S7**

**Table 4**  External drivers (i.e. socio-economic changes) of integrate modeling framework

| Yearly growth rate (%) | Stage 1 (2016~2020) | Stage 2 (2021~2030) | Stage 3 (2031~2040) |
|---|---|---|---|
| Population | 1.23 | 3.41 | 1.24 |
| Secondary industry | 1.99 | 4.11 | 2.36 |
| Tertiary industry | 3.04 | 5.33 | 1.24 |

Data sources: MGGC, http://data.cnki.net

## 5. Results

### 5.1 Coevolution process of EEF nexus

The coevolution trajectories of population, GDP, water supply & demand, streamflow, and objective function ($F_{ecnmy}$, $F_{eclgy}$, $F_{food}$, based on Eq.(1),(2),(3) and see **Table 2**) referring to each component of the EEF nexus is shown in **Fig.8**. As can be seen in **Fig.8**, the coevolution process of all the items depicts the characteristics of different stages. Finally, the (quasi-)stable state is converged, i.e., the variations of each variable are small or close to zero. It happens because the rate of external changes in the last stage (i.e., economic indexes) is much lower than in the previous stage, which decreases the internal changes (i.e., Streamflow water and three objective functions). Finally, the stable status of the whole system is achieved. In the first stage, the growth rate is relatively low and is based on the historical data, and the growth rate of $F_{ecnmy}$, $F_{eclgy}$, and $F_{food}$ is also slow. When entering the second stage, the economic growth will increase sharply to ensure the local economic development, and water demand is also increasing. However, according to the achievement of sustainable development based on the optimal model, ecological concerns should not be neglected. Therefore, the increase of river streamflow will also happen driven by the optimal model to maintain the river ecological health, consequently reducing the total water supply and increasing the water shortage of water users (**Fig.8**c). As $F_{food}$ and $F_{ecnmy}$ can reflect the water shortage of the corresponding water users, their value will also increase sharply (**Fig.8**e and **8g**) due to the rapid increase of socio-economic indexes. When entering the last stage, the development of socio-economy will tend to stable, and the increasing speed of $F_{food}$ and $F_{ecnmy}$ will decrease compared with that in the second stage. It is easy to understand because the relatively stable development of socio-economy does not need too much increased streamflow water (i.e., the increase rate of streamflow water is closed to a relatively stable state), and both changing rates of water supply and demand tend to be stable consequently (**Fig.8**c).

We can also see that the water supply system competes for the instream ecological system. As shown in **Fig.8**, especially in stage 2, increased streamflow is accompanied by increased $F_{ecnmy}$ and $F_{food}$ (**Fig.8**e and **8g**), reflecting the decreased satisfaction degree of the water supply of socio-economy and agriculture, thereby revealing the competition use of instream and off-stream water uses. The tradeoff between instream and off-stream water users can be obtained by the optimal model to solve for the best coordination status between them by adjusting economic development modes and balance the priority of each water users. It should be noted that the ecological objective ($F_{eclgy}$) is in a relatively stable status in all stages compared with other objectives (**Fig.8**f). This is because the ecological agent contains not only river streamflow but also vegetation. The booming economy drives the optimal model to focus more on river ecological health ($F_{riv}$), and there are limited water resources for off-stream water users including vegetation. The dual effect of increasing streamflow water and decreasing water for vegetation makes the $F_{eclgy}$ relatively stable. However, the optimal model takes the effect that the optimal allocation scheme is obtained by



shifting streamflow water because instream and off-stream water use is intrinsically conflicted with each other, and should be coordinated by adjusting different weights of each component (see section 6).

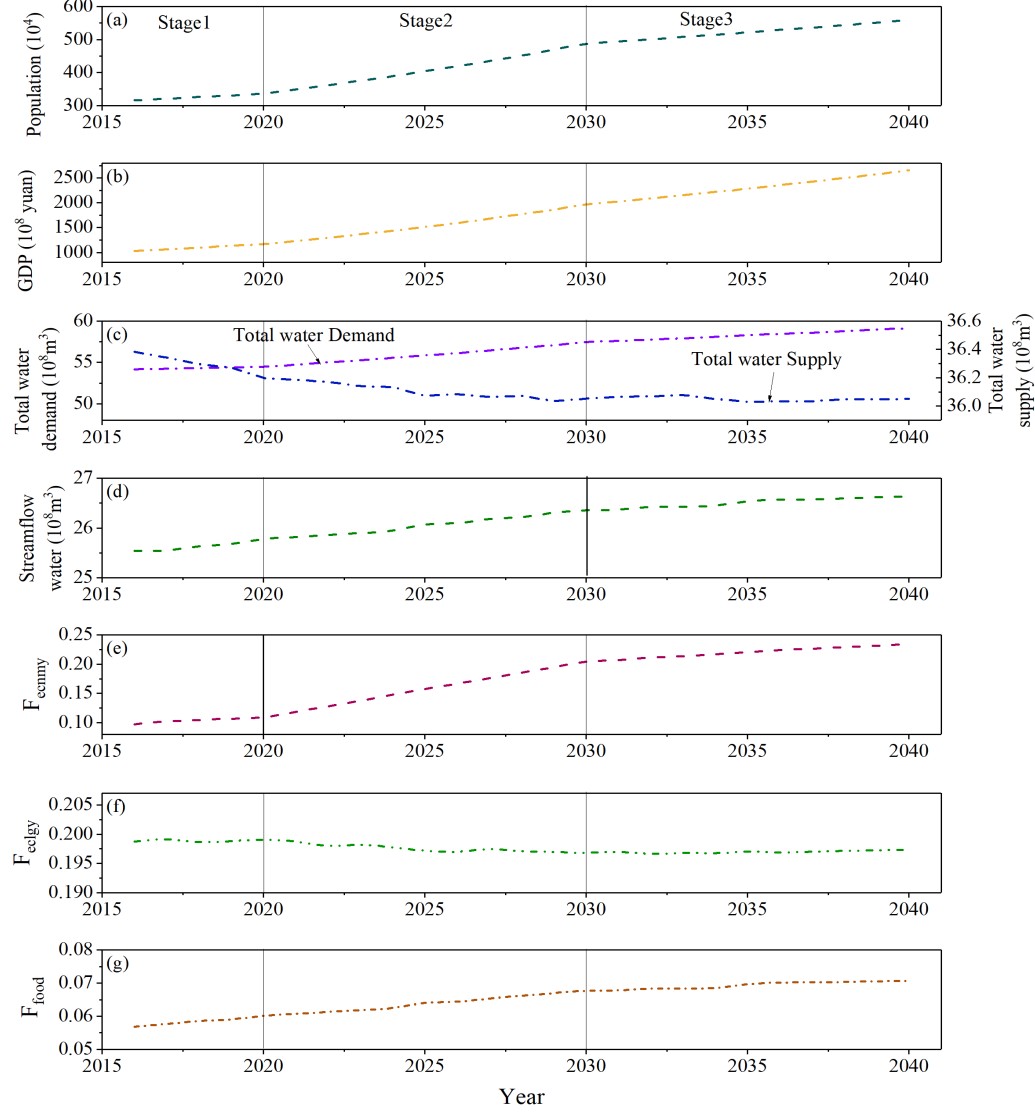

**Fig.8** Coevolution process of EEF nexus model

*5.2 Performances of feedback loops and response linkages*

*5.2.1 Economic-ecology response linkages*

  **Fig.9** illustrated the loop of economy-ecology feedback. As demonstrated in **Fig.9**, the response linkage of carrying capacity and overload index involves the changes of economic indexes, water supply & demand, and streamflow water (Feng et al., 2019). In the beginning, the economy is still increasing slowly, and the increasing rate





of water demand is also slow. The population and GDP are near the carrying capacity in this stage (i.e., the value of
       OI is near 1). In the following stage, both increasing population and GDP intensify the water demand (**Fig.9a** and **b**).
       To satisfy socio-economic development demands, water supply of economic agent has also increased. However,
       according to the coevolution of the whole system obtained by the optimal model, there will be a more significant
       concern of the river ecological system (**Fig.8c**, **Fig.9c**). In this view, the feedback linkage will take effect as that the
growing rate of water supply of household and industry (**Fig.9**d) will miss the rate of water demand (**Fig.9**b) and
       therefore contributes to the increase of water shortage, which is in accordance with the performance shown in **Fig.8**e.
       The increasing water shortage will generate the gap between carrying capacity (**Fig.9**e) and predicted economic
       indexes (**Fig.9**a). Then, the overload index will further increase, consequently affecting socio-economic development.
       It will force the local policymakers to readjust the regional development level and influence the population and GDP,
indicating a new round of feedback. In this view, we can see that the rapid growth of economy in the second stage
       will activate the protection mechanism of river ecology by increasing the streamflow, and the rest water is unable to
       support the increasing economic development. It further contributes to the overload of the water resources system,
       which even restricts the socio-economy instead. In the last stage, the continuous increase of the overload index
       stimulates the policymakers to alleviate the growth rate of population and GDP (**Fig.9a** and **f**). It forces the relatively
slower increase rate of streamflow water, and there will be more water space for socio-economic development.
       Although the water shortage is increasing, its rate is lower than that in the second stage. The carrying capacity will
       be able to catch the predicted economic index if the stable or slower growth rate continues. The overload index is
       also decreased, and the whole system tends to be stable. Thus, this is the negative feedback loop (see Section 3.2.2)
       that eventually makes the system stable, reflecting that the rapid increase of the population will finally readjust the
economic index itself by triggering the awareness of streamflow.



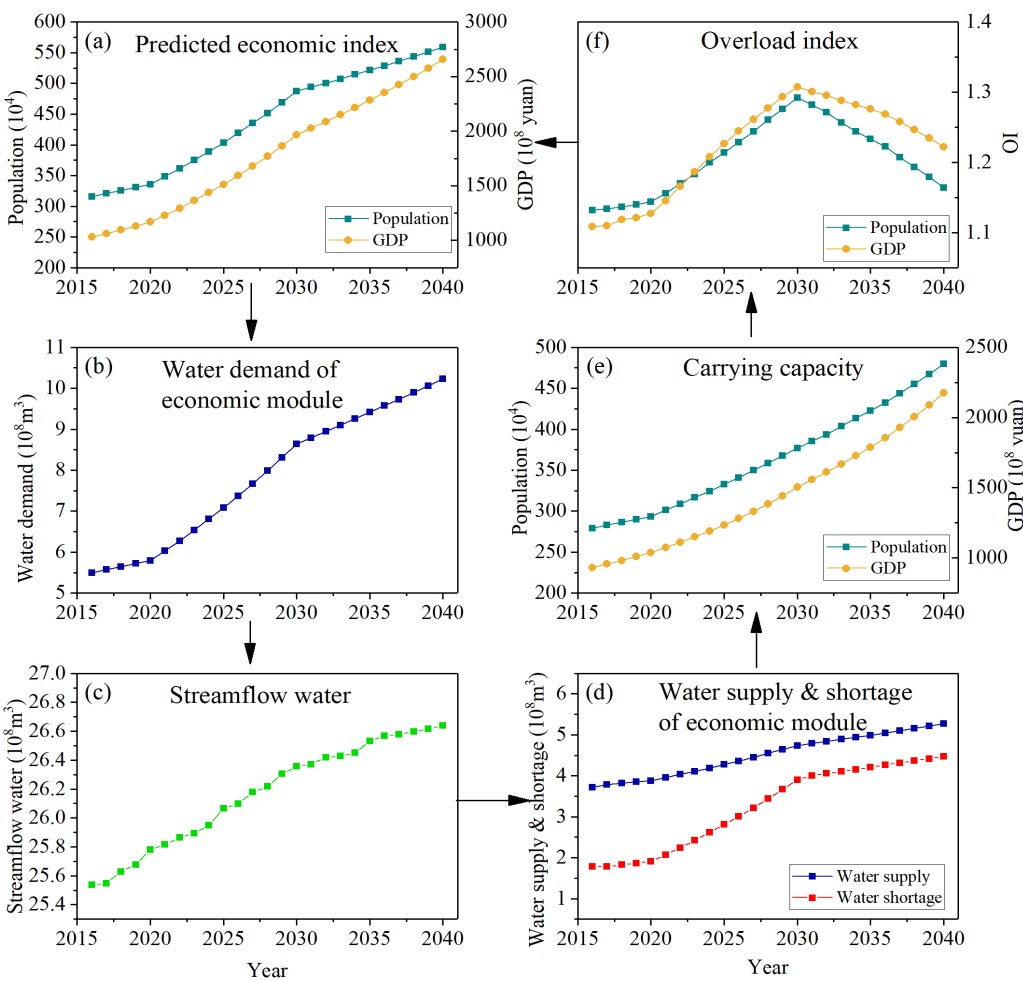

**Fig.9** Response linkage of economy-ecology feedback loop

### 5.2.2 Ecology-food response linkages

Another performance is the ecology-food response linkage and is shown in **Fig.10**. It not only illustrated the

linkage between food and ecological water usage but also demonstrated the coevolution of ecology components of
both instream (river ecology) and off-stream (vegetation) aspects. We can see from **Fig.10** that the increased
streamflow water is the driving force of ecology-food response. However, the increasing streamflow water was driven
by the rapidly increasing socio-economic scale. The optimal model is used to achieve the goal of sustainable
development to balance the need of different users, especially that of instream and off-stream. The increased

streamflow has two effects in ecology-food response linkage. First, the variable $F_{riv}$ describes the ecological health
of a certain river. According to the definition of AAPFD, the higher value of streamflow water indicates the lower
value of $F_{riv}$, which indicates that the river ecology is getting better. Second, the increasing streamflow water restricts
the water supply of all off-stream water users, including agricultural and vegetation water (**Fig.10**b). Irrigation and





vegetation water use is the largest off-stream water consumer, and their increased water shortage was also driven by
increased streamflow water (**Fig.10**d). It should be noted that the food agent includes not only crops but also livestock.
Livestock breeding will inevitably increase to make more production value of the primary industry, and there will
consequently be more water demand for livestock.

**Fig.10**   Ecology-food response linkage

540        The dual effect of increased streamflow water and decreased vegetation water makes the stable change of $F_{eclgy}$
(**Fig.10**e), indicating that the ecological aspect of UGRB is maintaining a good status. Since crop yield is strongly
affected by the satisfaction degree of irrigation water, and the increased water shortage of crop water will, therefore,
indicate the decrease of crop yields (**Fig.10**f). In contrast, the decreased water shortage of livestock could induce an





increase in meat production. The detailed changes in crop yield and meat production are presented in **Fig.11**. We
can see from **Fig.11** that a relatively large proportion of food production is from crop yield. Although meat
production is increasing, it accounts for relatively less proportion, and thereby the total food production will first
decrease and then tend to be stable in the last stage (**Fig.11**c).

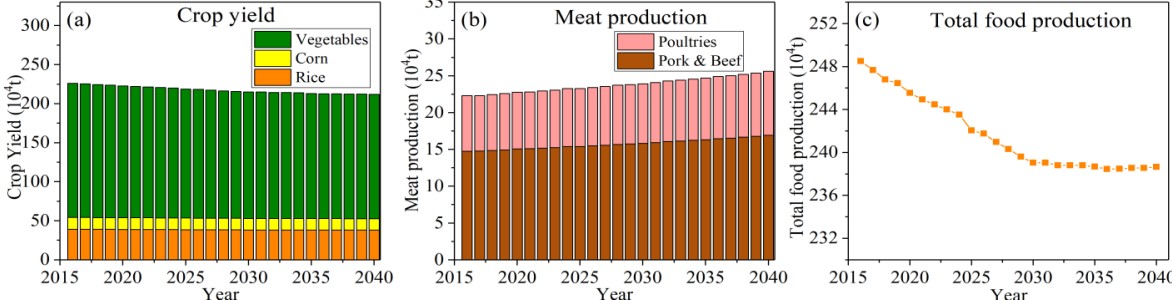

**Fig.11** Detailed crop yield and meat production in the next 25 years in UGRB

*5.2.3 Economy-food response linkage*

The decreased food production is driven by the increased streamflow water that also caused an increasing
overload index (**Fig.9**f) in the second stage, and it is reflected by economy-food response linkage (**Fig.12**). The
decreased food production also results in the stagnant farmer's income. This happens because of the dual effect of
both increased food production and increased population. Food production is considerably related to the total value
of the primary industry. The reduced crop yield increases the food price, but its rate is still less than the rate of
population growth. Therefore, the stagnant income and decreased crop yield will finally decrease carrying capacity
and further intensify the overload index. It is easy to understand because food and income is the most fundamental
substance of people's survival. If the growth rate of the predicted population decreases, there will be less pressure for
water supply and can well balance the agricultural and streamflow water, further contributing to stable food
production, increased farmer's income, and decreased overload index. So far, the linkage of economy-food, economy-
ecology, and ecology-food were all presented, which indicated that the three components interact and respond with
each other.

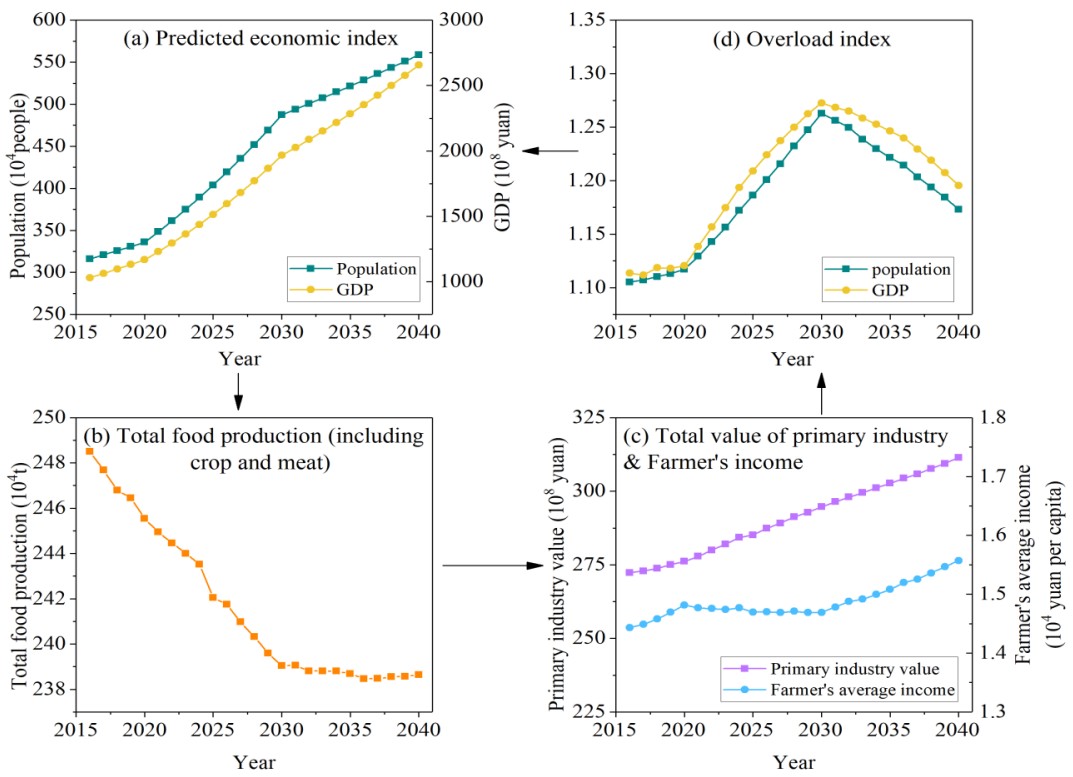

**Fig.12** Economy-food response linkage

*5.3 Assessment of coordinative degree of each subsystem and SDD*

The calculation result of SDD of EEF nexus and coordination degree of the economy (ECNMY), ecology (ECLGY), and food (FOOD) is demonstrated in **Fig.13**. We can see that the variation of the four variables is also showing the state characteristics. The ECNMY in the first stage is increasing, but it had an either decreasing (UGRB, Guilin urban area, Lingui, etc.) or stable (Xing'an, Yangshuo) trend in the second stage, indicating the coordinative status of socio-economy is not good caused by the excessive growth rate of the economy. The decreased coordinative status of the economy subsystem also influences other subsystems and the SDD of total EEF nexus, reflected by the decrease of ECLGY, FOOD, and further SDD. Fortunately, the decreasing rate of ECLGY is smoother compared with that of FOOD, indicating the performance of the ecology of UGRB is relative well compared with socioeconomics and agriculture. This performance could be due to the dual effect of increasing streamflow water, sewage and recycled water treatment, and decreasing vegetation irrigation. The same was true for other administrative regions of UGRB. Moreover, for the whole basin, the value of ECNMY in the later period of the second stage (about 2028~2030) is even lower than FOOD and ECLGY. From the perspective of administrative regions, it is more obvious in Guilin urban area, Pingle, and Lipu counties. It happens because the economic-stressed stage has been lasted almost ten years in 2030, which is similar to the "pendulum model" that takes the effect that the pendulum "swing" towards the economic-stressed system (See 2.1.1). As socio-economic index increases sharply and continuously, the



ecological protection mechanism will also be continuously triggered to increase the overload index, resulting in both ECNMY and SDD reached the minimum.

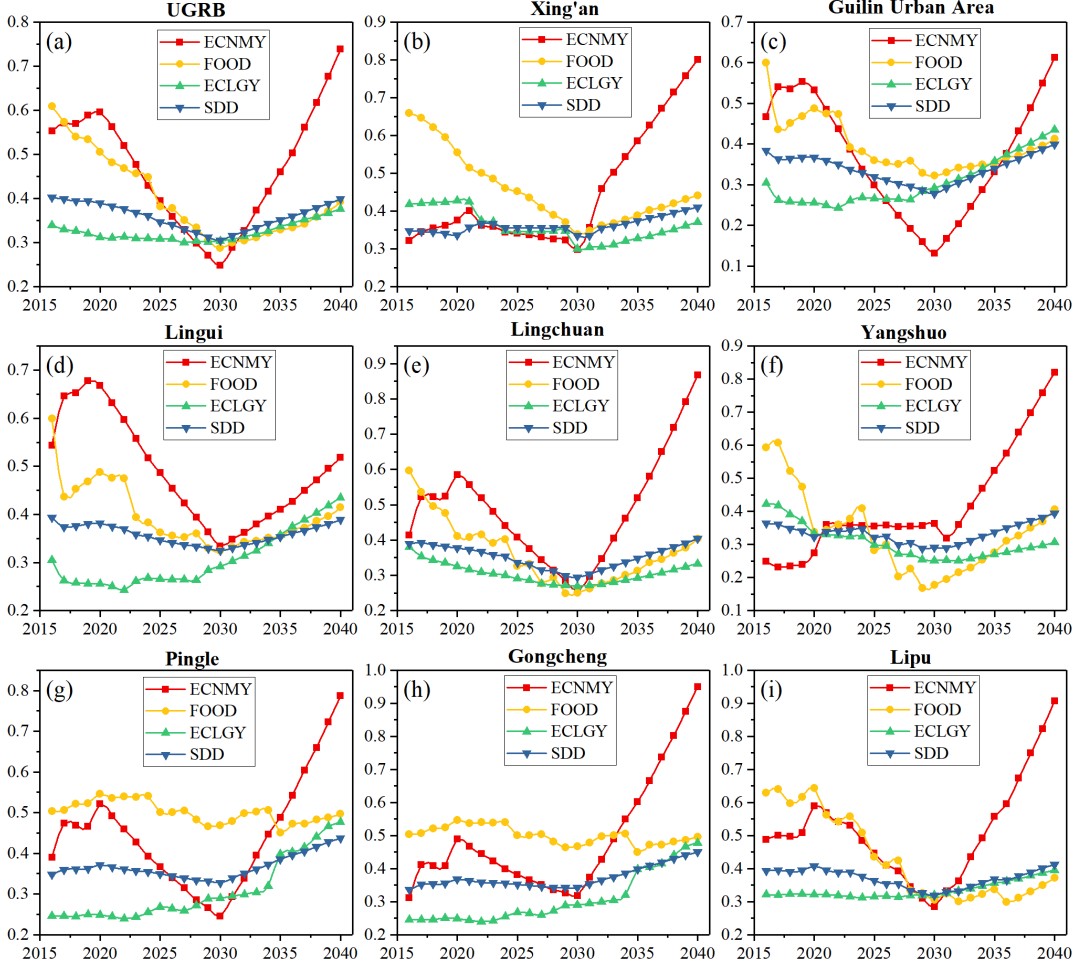

**Fig.13** Time variation of sustainable development degree (SDD) of EEF nexus and coordination degree of each agent

When it comes to the third stage, the value of ECNMY increases, indicating the coordination of the economic subsystem is improving. It revealed the decreasing of overload index and the increased carrying capacity due to the relatively slower increasing rate of water demand of economic agent. The increasing value of ECNMY even promotes the coordinative degree of ecology and food, and the value of SDD is consequently increased, revealing that the stable economic growth will promote the sustainable development of EEF nexus. The good phenomenon of the last

stage happens because the relatively slow growth rate of water demand for the economic agent will generate more water for food and ecology, and the increasing sewage and recycled water treatment rate will provide relatively more water for users. The coevolution process assumes the "pendulum model" presented by Van et al. (2014) and Kandasamy et al. (2014), where environmental awareness has been raised, and a stable population rate occurred in the last era. The result presented in this study is similar to the findings in Van et al. (2014) and Kandasamy et al.





(2014). Furthermore, we can speculate that in the 2040s, the pendulum of ULRB will also "swung" back to the stage of protective resources & environment and stable development of socio-economy, just as stated in Kandasamy et al., (2014).

## 6. Discussion

### 6.1 Decision making performance considering model uncertainty

The chain of the model is complex and usually contains lots of uncertainties. In this case, decision-makers usually aim to achieve multiple performance objectives and have to make tradeoffs among those conflicting objectives, which arises from uncertainties (Herman et al., 2015). Overall, the procedure of uncertainty analysis includes its identification, tradeoff evaluations, and selection of the best solution for stakeholders (Herman et al., 2014). The source of uncertainty usually derives from both external and internal aspects. The external aspects are 605 reflected by the social development degree, climate change, natural hazards, etc. In contrast, internal aspects are reflected by system model designation, particularly for systems with multiple stakeholders. In this case, optimal modeling approach is used to solve such a system model problem based on the multi-objective optimal theory that refers to the tradeoffs between multiple stakeholders. As multi-objective optimization problems cannot give one solution, it provides the non-dominant possibilities, that is, the solution portfolios, and is usually called Pareto frontier. 610 It is commonly used in resources distribution issues based on multi-objective nexus systems, and each one of the solutions out of the portfolio is all possible solutions that are not inferior to all other feasible solutions (Woodruff et al., 2013; Hadka and Reed, 2013). Meanwhile, Herman et al. (2015) addressed the taxonomy of robustness frameworks that includes identifying alternatives, quantification of robustness, and controls based on Many-objective Robust Decision Making (MORDM) for uncertainty analysis (Kasprzyk et al., 2013). A set of discrete alternatives 615 can be prespecified by decision-makers and is the fundamental method of uncertainty analysis (Herman et al., 2015). Therefore, how to choose those optimal solutions from the Pareto portfolio is the main source of the model uncertainty by which the weight of each objective is reflected (Tingstad et al., 2014; Liu et al., 2019). In other words, it is the tradeoffs across objectives.



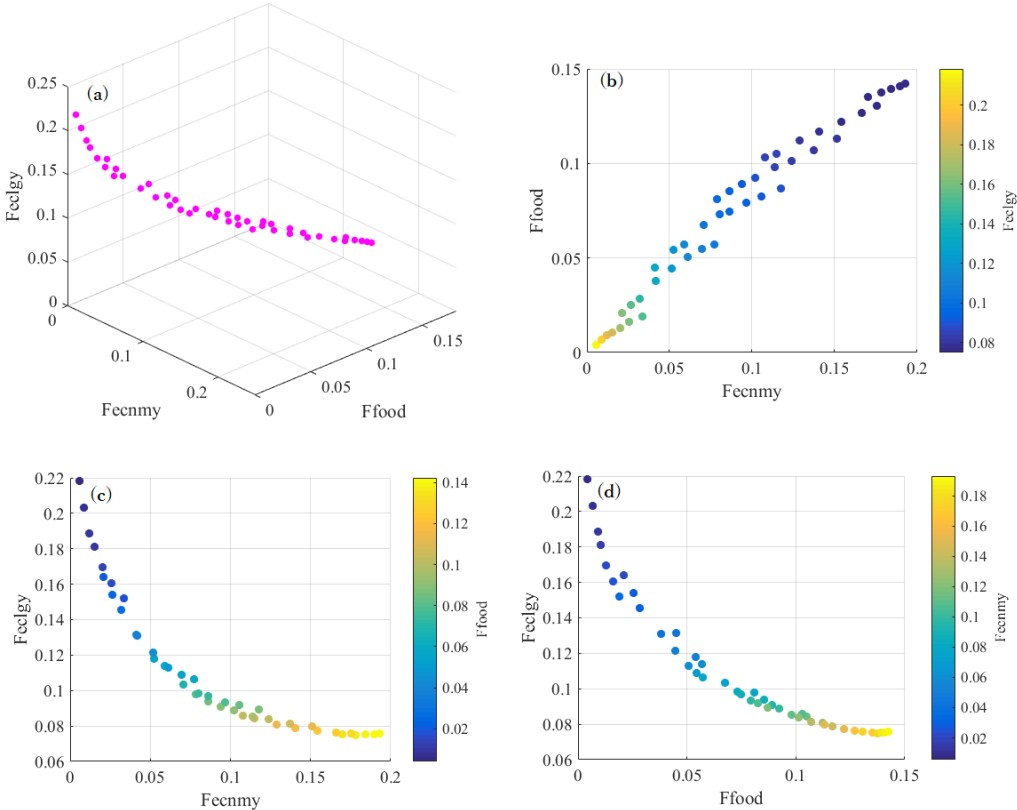

**Fig.14** Competitive mechanism of EEF nexus

The Pareto-optimal solutions in 2016 are demonstrated to show the competition mechanism and tradeoff across the objectives of the EEF nexus (**Fig.14**). The Pareto solutions of other years show similar performance, and it will be redundant to demonstrate all the results. It is set just an example to illustrate the tradeoff mechanism. **Fig.14**(a) shows the non-dominated (or Pareto) optimal solution of EEF nexus, and suggested that the maximum benefit of one
of the three objectives should be in exchange of sacrificing either of the other two objectives. It reflects the competitive linkages among socio-economic development, food production (or farmer profit), and environmental protection. **Fig.14**(b)~(d) shows the detailed competition relationship among objectives and can be stated as (1) The mutual relationship among three objectives are nonlinear, (2) both $F_{ecnmy}$ and $F_{food}$ increases as $F_{eclgy}$ decreases, and (3) either the reduction in $F_{ecnmy}$, or $F_{food}$, will increase $F_{eclgy}$. The latter two indicate the conflict between instream
water use ($F_{eclgy}$) and off-stream water use ($F_{ecmny}$, $F_{food}$).

Since each dot in Pareto frontier is the optimal solutions that correspond to a certain weight vector $r=(\alpha_1,\alpha_2,\alpha_3,\theta)$, where $\alpha_1,\alpha_2,\alpha_3,$ and $\theta$ respectively represent the weighting factors of economy, vegetation ecology, food, and river ecology, the uncertainty arises because decision-makers are usually hard to choose which one is the better than another one. Thus, according to Herman et al. (2014 & 2015), selecting the best solution is the key procedure of
uncertainty assessment because it is based on multiple scenarios or states instead of a single probable future scenario. The optimal solution portfolio is the ensemble of the internal uncertainty factors, and the uncertainty assessment





should include the constructing process of several discrete alternatives that may be available for various stakeholders by sampling those uncertain factors (Bryant and Lempert, 2010). Therefore, this study provides several alternatives based on different weighting factors to assess model performances. Twelve alternatives are presented in Table 5 and

represent the preferences of decision-makers, and the different performances are shown in Fig.15.

Table 5   Twelve alternatives based on weighting factors for uncertainty assessment

| Alternatives | Weighting factors | | | | Alternatives | Weighting factors | | | |
|---|---|---|---|---|---|---|---|---|---|
| | $\alpha_1$ | $\alpha_2$ | $\alpha_3$ | $\theta$ | | $\alpha_1$ | $\alpha_2$ | $\alpha_3$ | $\theta$ |
| A1 | 0.2 | 0.1 | 0.2 | 0.5 | A7 | 0.2 | 0.2 | 0.4 | 0.2 |
| A2 | 0.2 | 0.1 | 0.3 | 0.4 | A8 | 0.5 | 0.2 | 0.1 | 0.2 |
| A3 | 0.2 | 0.2 | 0.2 | 0.4 | A9 | 0.4 | 0.2 | 0.2 | 0.2 |
| A4 | 0.1 | 0.2 | 0.4 | 0.3 | A10 | 0.5 | 0.1 | 0.2 | 0.2 |
| A5 | 0.2 | 0.1 | 0.4 | 0.3 | A11 | 0.4 | 0.1 | 0.2 | 0.3 |
| A6 | 0.3 | 0.1 | 0.4 | 0.2 | A12 | 0.25 | 0.25 | 0.25 | 0.25 |

As stated in Section 3.1.3, each weighting factor represents the preference of different decision-makers. Each $r$ is called alternative, and therefore twelve alternatives (A1, A2, …, A12) are presented. Approximately, A1 to A3 focus more on ecological streamflow with higher $\theta$, while that of A4~A7 and A8~A10 is lower. A4~A7 focus more

on food agent while A8~A11 focus more on economic agent. A11 focuses on both economic and streamflow issues. A12 is the average level that each weight is set as equal. The value of both objective function of each agent and SDD under each alternative is shown in Fig.15. Generally, the values of SDD under A1~A5 and A11 are smaller than those under other alternatives. Meanwhile, the objective function of both economy and food agent under A1~A5 and A11 is higher than that under other alternatives, suggesting the more water shortage. On the contrary,

the objective function of ecology agent shows the opposite trend. We can contribute this result to the relatively higher weighting factor of $\theta$ and the lower weighting factor of $\alpha$ in those alternatives, resulting in the relatively less water serving for economic and food agents, and finally cause more water shortage of these agents. Moreover, of all the alternatives, A12 performs the best, suggesting that equal consideration to each agent is more likely to attain sustainable development. The parameter $\theta$ and $\alpha$ equal 0.25 in A12, while the value in other alternatives is either

more or less than 0.25, suggesting that excessive or lower weighting factors prevent the sustainable development of water resources to some extent. Therefore, the uncertainty analysis can also give a strong reference for the decision-making process for water resources management.



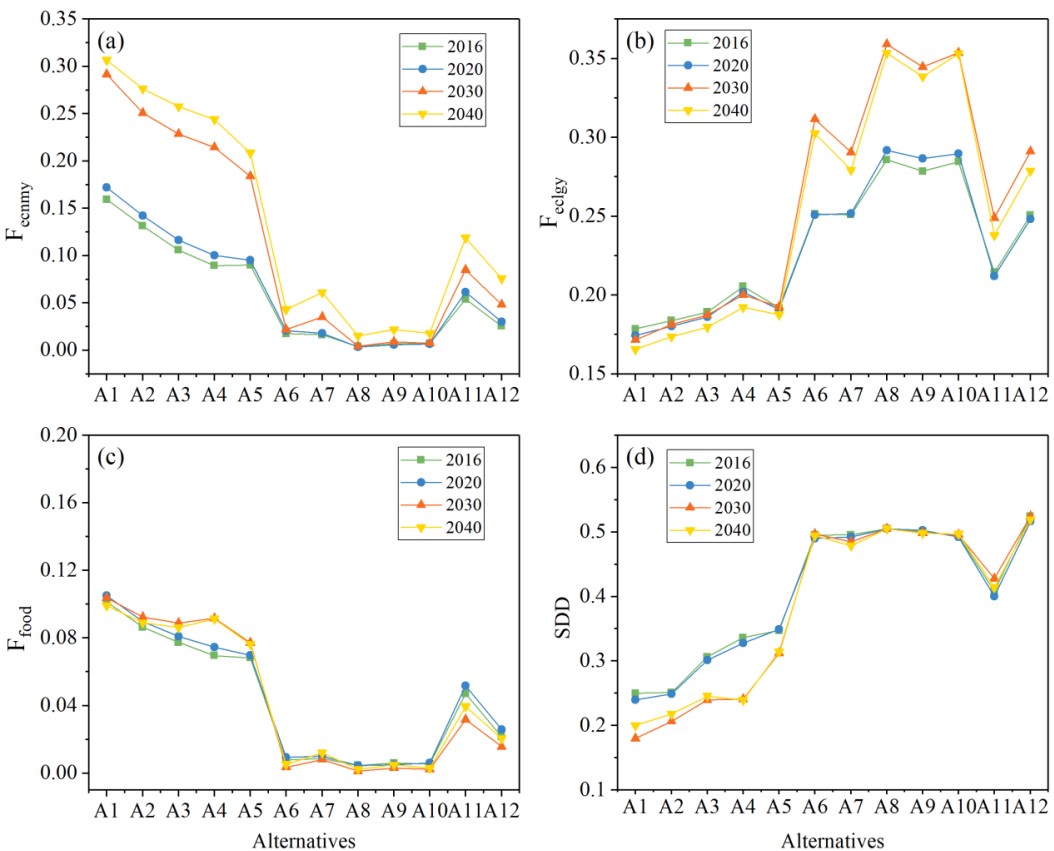

Fig.15   Sustainable development degree of different alternatives

*6.2 Robustness analysis for EEF nexus*

The key factor(s) that affect the robustness of the EEF nexus system is/are assessed to improve its reliability. The alternatives of A5, A7, A9, A11 are set particularly by controlling relative variables to assess the robustness of EEF nexus. In the case of both A5 vs. A7 and A9 vs. A11, we change $\theta$ while $\alpha_1$ and $\alpha_3$ remain unchanged to assess the influences of river ecology water changes on the performance of EEF nexus. While in the case of both A5 vs. A11 and A7 vs. A9, we change $\alpha_1$ and $\alpha_3$ while $\theta$ and $\alpha_2$ remain unchanged to assess the influences of water changes of both economic and food agents on the performance of EEF nexus. According to Fig.15, the differences between both cases are shown in Table 6. To illustrate, the SDD value of 0.06 in row "A5 vs. A11" and column "2016" means that the difference of SDD value between A5 and A11 in 2016 is 0.06. From Table 6, we can see that the values in the lower two rows are smaller than those in the upper two rows. It indicates that when the weighting factors of both economic and food agents are certain, changing the weighting factor of streamflow will have a relatively significant impact on the performance of EEF nexus in both objective function and sustainable development degree. Additionally, changing the weighting factor of both economic and food water uses will have less influence on model performance. In other words, the streamflow agent has a relatively great influence on the robustness of the EEF nexus model.



Table 6 Comparison of the performance of EEF nexus between different alternatives

| Case comparisons | Uses | $F_{ecnmy}$ | | | | $F_{food}$ | | | | SDD | | | |
|---|---|---|---|---|---|---|---|---|---|---|---|---|---|
| | | 2016 | 2020 | 2030 | 2040 | 2016 | 2020 | 2030 | 2040 | 2016 | 2020 | 2030 | 2040 |
| A5 vs. A7 | Influence of changing river | 0.07 | 0.08 | 0.15 | 0.15 | 0.06 | 0.06 | 0.07 | 0.06 | 0.15 | 0.14 | 0.17 | 0.16 |
| A9 vs. A11 | ecology on EEF performance | 0.05 | 0.06 | 0.11 | 0.12 | 0.04 | 0.05 | 0.05 | 0.05 | 0.09 | 0.10 | 0.12 | 0.12 |
| A5 vs. A11 | Influence of changing ecnmy | 0.04 | 0.03 | 0.08 | 0.06 | 0.02 | 0.02 | 0.02 | 0.02 | 0.06 | 0.05 | 0.07 | 0.07 |
| A7 vs. A9 | and food on EEF performance | 0.01 | 0.01 | 0.03 | 0.04 | 0.00 | 0.01 | 0.00 | 0.01 | 0.01 | 0.01 | 0.01 | 0.02 |

The robustness of river ecology can also be reflected in the model performance of different years. From Fig.15, we can also see that both objective functions and SDD under A1~A5 have a greater difference between 2016&2020 and 2030&2040 compared with other alternatives. There will be a rapid economic increase from 2020 to 2030, and the ecological awareness in these alternatives outweighs other alternatives (with higher θ), which is more likely to trigger the adaptive adjustment of the complex system and further accelerates the river streamflow. Then, there will

be not enough economic water services, and the overload index will increase, further decreasing SDD in 2030 compared with 2020.

*6.3 Simplifications of model dynamics and limitations*

   The proposed model simulates the dynamic evolution and feedback loops based on the three agents: economy, food, and ecology. The result proposed in this study is quite similar to Kandasamy et al. (2014) because he stressed

that environmental awareness arises when an accelerated population is about to consume freshwater and results in the decrease of the population to protect the environment. This study also proposed the stable status of sustainability of both social productivities and environmental issues because the population growth rate is moderate and steady in the third stage to pay more attention to environmental awareness.

   These individual three items are also prominent aspects or disciplines that contain numerous basic principles.

Therefore, several assumptions and simplifications are often conducted to develop the nexus models that are, to some extent, one of the most necessary and significant ways for natural resources management practices for sustainable development. For example, food production and primary industry that belong to agricultural productivities are determined by the original external conditions: precipitation and potential evapotranspiration that belongs to climate. The long-term historical climate data is used for the input of the model based on the assumption that the long enough

historical data (monthly or even daily, several decades) can represent all possible climate scenarios. Not only EEF nexus, but also other nexus is also based on several assumptions that are not always perfect. For example, the WPE (water-power-environment) nexus developed by Feng et al. (2019) considered water use quotas based on the exponential assumption. Still, they ignored the dependence on population growth rates.

   The above two examples are purposed to illustrate that several assumptions should be often conducted before

developing a certain model, which is also a key procedure of most scientific researches. But some assumptions may ignore some of the basic principles and further limit the models (Pindyck, 2015). For example, for ecological aspect, it not only includes vegetation and streamflow but also consists of the water ecology and the issue that the human activities may lead to the ecological damages (Factories may increase $CO_2$ emissions), as well as the pollution from agricultural productivities. The population and GDP growth, to which the water demand of economic agents is related,

is calculated by the Malthusian model (**see Supplementary material S2**) that may work only on a short time scale (about 2~3 decades). Still, it may not hold for a long time (100 years for example) scale. Those are all the model





limitations that can be considered in our future research since the current study has simplified some of the basic principles as noted above. Therefore, no model is absolutely accurate and perfect, and several assumptions should be considered, although those assumptions sometimes are not that adequately (Pindyck, 2015). However, assumptions

are the necessary procedures in most scientific researches. Thus, modeling approaches must be developed to solve a certain problem. Despite the limitation stated before, it does not mean that it is of no use or to give up entirely on estimating the sustainable development status more generally (Pindyck, 2015). We need to take advantage of the positive effect as much as possible of a certain model that is, although, usually double-sided.

## 7. Conclusions

This paper presented a new integrated framework that is used to analyze the dynamic interactions within coupled human and natural systems in the context of socio-economic development, food safety, and environmental protection by establishing integrated and systematic modeling. The framework revealed the dynamics by introducing complex adaptive system theory, and the adaptive process of each agent under changing external conditions is attained through system optimization model and system dynamic model in an interactive and dynamic way, capturing the dynamic

coevolution and feedback loop of the integrated system and produced the trajectories of future changes. The changing external conditions, i.e., the socio-economic development changes, result in nonlinear and multiscale feedback responses. The coevolution process of social and natural systems, including economic, food, and ecology agents, is generated in complex and interactive ways. The uncertainty analysis is also helpful for multiple tradeoffs and robustness analysis in the decision-making process. The result can give a firm reference and provide a practical tool

for water management and policymakers from the following aspects:

This coupled modeling tools enable the coevolution and feedback loop dynamically by generating the whole scale of future trajectories that reveals the interactions across economic development, food safety, and ecological protection. In this paper, the feedback loops between any of the components of EEF nexus under different stages of economic levels are explored. All the trajectories differed in different stages. There are no obvious changes in the

performances of the model in the first stage. In stage 2 (2021~2030), the severe increase of economy intensifies the interaction of a complex system, triggering the more streamflow water of reservoirs for ecological agent. It results in less water for agriculture and social economy and cannot afford the rapidly increasing population, economy, and food yield. In stage 3 (2031~2040), with respect to moderate development of socio-economy, the interaction of the nexus system will be alleviated, that is, the increasing rate of streamflow water will be decreased, and there will be more

water to support the population size and economy. The results suggest that only considering the economic benefits will rather accelerate the overload process of the overall system, which inversely affects the socio-economic development. It also shows that, depending on the external drivers, the dynamic changes manifest differently in water supply, streamflow water, farmer's profit, and population size. Thus, the coevolution process and reciprocal feedbacks between human society and natural systems can provide valuable information and guideline for policymakers on how

to decide the development degree and manage water resources on a regional scale considering economic development, food safety, and ecological protection.

The uncertainty analysis result of the coupled model also revealed the different performances considering the need of various stakeholders, giving references to multiple tradeoffs influencing integrated systems and stakeholders, notably the tradeoffs between water for social development, food production, and ecological protection. The Pareto

portfolio of the multi-optimization model based on different weighting factors reveals the competitive mechanism of the three agents of the coupled model. The alternatives based on different weighting factors show the varied





sustainable development degrees and objective functions of each agent. Of all the alternatives, the equal consideration of each stakeholder (weighting factor) is more likely to achieve sustainable development. Therefore, policymakers can explore the future water allocation scheme among different needs of stakeholders based on those different alternatives. Of all the agents within the integrated system, the river ecological part is more likely to influence its robustness. This result suggests that the ecological agent of the integrated water resources system should be paid more attention to the process of both water allocation and the policymaking process. The integrated modeling framework presented in this paper is designed to simulate the interactions and feedback responses across multiple agents, and the uncertainty analysis can improve the model reliability to provide valuable predictive insights into the decision-making process of integrated systems.

**Acknowledgements**: The project was financially supported by National Key Research and Development Program of China (No. 2018YFC1508200), National Science Foundation of Jiangsu (No. BK20181059) and China Scholarship Council. The authors were also grateful to the sources of hydrological and meteorological data from hydrological authority and statistical bureau, and the organizations and comments handled by Dr. Zengchuan Dong and Dr. Sandra M. Guzman. The authors are still grateful to the insights and views of the editors and reviewers.

## Supplement: Supplementary materials (Data availability)

(Supplementary materials, uploaded to the supplementary links of journal's website)

## Author contribution

Yaogeng Tan prepared the manuscript and developed the model. Zengchuan Dong and Sandra M Guzman revised the manuscript. Sandra M Guzman also helped developing the model. Xinkui Wang and Wei Yan helped collect the data.

## Competing interests

The authors declare that they have no conflict of interest.

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
