# Peer review of "Modeling the integrated framework of complex water resources system considering economic development, ecological protection, and"

_Hydrology and Earth System Sciences, 2020_

## Referee Comment (RC1) · Anonymous Referee #1 · 26 Oct 2020

General: This paper develops the complex water resources system considering three elements: economic development, ecological protection, and food safety (namely EEF nexus as authors state in the text) that is the most likely to consume water resources and applied in a case study of Guijiang River Basin. The framework is of novelty, and the authors disclosed the optimal system dynamic and coevolution & feedback results by coupling the optimal model and SD model and found that only considering economic development will result in the overload status of the entire system. Once the ecology is paid attention to, sustainable development status will be achieved. The authors also

considered the model uncertainty and sensitivity to make the model reliable. Generally, the paper is good, with an interesting topic, and has a great significance for water resources management communities, and it also well fits the scope of HESS. However, I found two main flaws in this paper that must be paid attention before it can be considered for publication.

Major concerns:

Firstly, the logic of the text should be improved. In other words, the relationship between SD and optimal model, and its combination is not clear or, difficult to understand. This is the main flaw of this text. The description of both individual models is well-explained but their relationship is unclear. The authors outlined the general framework of the EEF nexus (shown in Fig.1) but in the following section, it suddenly jumped to the optimization model before introducing the SD model, which is illogical to some extent when reading this paper and difficult to understand. What is the relationship between these two models and how to connect with each other? The framework of the EEF nexus itself is exactly the reflection of the SD model and, in my opinion, if the optimal model is interrupted within the text, the logic will be damaged. Generally, the results of SD model are not always optimal if not considering the multiple water use agents because optimal algorithms are inherently used to tackle multiple objective problems. SD model is used to simulate the real-world status of a certain system that is composed of both variable relationships and their positive/negative feedbacks but has no optimal functions. Thus, the results of some variables are not always optimal. Based on this, the optimal model should be then applied to SD to generate the optimal result, and then the dynamic results considering the external temporal changes will be proposed (Section 5). That's why we should couple two models to deal with the research problem in this paper. So far the authors did not realize this logic problem and the logic of the text is flawed for me. To overcome this problem, I suggest that section 3.2 (SD model, at least 3.2.1 and 3.2.2) should be explained before the optimal model (section 3.1) to better explain the logic of the text. In other words, replace the order of both models, and the

research framework (Fig.2) should also be adjusted to reveal the logical relationship. Otherwise, the optimal equations and SD are feeling disconnected as the editor noted before interactive discussion. Although the editor mentioned the links between equations and SD should be provided, I would still stress the above-mentioned aspects to better connect the optimal and SD model.

Secondly, the calibration and validation process of the SD model should be supplemented and I also concern about the time series. The authors stressed three different develop levels (2016-2020, 2021-2030, 2031-2040) that corresponds to the pendulum model. However, it is 2020 already and I am worrying if the calculation of 2016-2020 is of significance. Also, SD model should be calibrated and validated before using it in a real case study but I cannot find any of the relative statements. In Table 4 the authors addressed the external drivers of three different stages. There's no doubt that stage 2 and stage 3 is the output of both model, but it remains unclear what the result of stage 1 represents, which is difficult for readers to understand. Is that the simulation result or it already exists in a real case? If it is the simulation result, it should be compared with the observed (or statistic) data for more model reliability, or otherwise, all the results will not be convincing. If it is the observed data, the authors should make it clear what the relationship between the results of stage 1 and stage 2 & 3 and how the dynamic runs overtime. Please clarify this issue in the revised paper.

Other concerns:

L26: In what way does the paper "highlight the importance of water resources management"? Please explain.

L85: are usually manifested.

L113: The framework is not yet raised to theoretical altitude. It is a little bit exaggerating. The word, such as "overall", "general" may be better.

L139-147: Why single out "CAS theory" here? Please explain the usage in this text

and the relationship of subsequent sections (or add some relative statements).

L150-158: As noted before, SD and optimal model should be reconsidered. Therefore, this part should be thoroughly edited to reveal the correct logic relationship.

L656-657: Please explain why uncertainty analysis "gives a strong reference for decision-making process".

L720-721: The general brief sentence should be added to explain how the research framework "becomes a practical tool". The following two paragraphs is used to disclose the general brief sentence. Also, check if the abstract has mentioned this. The title includes "practical tool" and all the text should be surrounded this issue.

I'm looking forward the revised version of the text.

---

## Referee Comment (RC2) · Anonymous Referee #2 · 24 Dec 2020

This paper provides a framework to study economic development, ecological protection, and food productivity (EEF) nexus by integrating optimization and system dynamics. The model presented in the paper predicts major performance measures of the nexus for the Upper reaches of Guijiang River Basin (UGRB) in China. Although I agree with the comments posted by the editor and the first reviewer, I believe the paper suffers from some more significant issues.

First, it is not clear what the contribution of the paper is. Is the paper trying to introduce a new method? Is it trying to solve a problem? Is it trying to introduce a new

theory? The paper needs to have a clear goal and then focus on that. If the goal is to introduce a new method, it should provide clear information on how optimization and system dynamics can be integrated. The current manuscript does tell us much about the integration process, as explained elegantly by the first reviewer. If the goal is to solve a problem, the problem should be defined. The current presentation provides some predictions about some measures and explains their dynamics, but it is unclear what problem it is trying to solve. Is the problem unsustainability? Is it depletion of resources? What are the research questions? What is the hypothesis that the paper aims to test? And then, it must provide some robust solutions. The paper has some scattered solutions in the discussion section, but they need to be organized and become the focus of the paper if the goal is to solve a problem. Finally, if the goal is to advance a theory, the whole writing should support that goal. The current manuscript aims to achieve all of these goals, and that makes it ambiguous.

Second, the paper is hard to follow mainly because its English writing is poor. There are many instances of grammatical and typing errors, as well as bad writing. I have listed some of the examples at the end of this note.

Third, the paper does not get the system dynamics terminology right. The method is mistakenly called "system dynamic," which is wrong. Another instance is the use of "systematics" as a misnomer of "systems perspective" or "systems relationships." Yet another example is the use of "cause-and-effect feedback loop" (Fig. 6 caption) instead of "causal loop diagram." Also, some statements in the manuscript make me nervous and skeptical about the authors' command of system dynamics modeling. They claim: "if the water supply increases at the same rate as water demand caused by increased socio-economic index, this feedback will be the positive feedback regulation that results in the polarization because the ecological water will be occupied and environmental protection will not be guaranteed." Note the use of "if" and "this feedback will be" in this statement. A feedback loop cannot be conditional; it is either positive or negative. The ambiguity of a feedback loop indicates a major deficiency in the model.

[Figure]

Fourth, the presented model is very confusing. It is not clear if the model is trying to "characterize" the real system or represent an ideal situation. If the goal is to characterize the real system, then a system dynamics model would be sufficient. Agents of a real system rarely optimize things. If the goal is to represent an ideal situation (in a neoclassical sense), then the optimization model would be sufficient. I believe this confusion arises because the goal of the modeling is not defined adequately. As the first reviewer explained, the relationships between the optimization and the SD model need to be clarified.

Fifth, the presented model has some questionable assumptions. For example, it assumes that "The goal of the economic agent is aiming at increasing revenue of secondary and tertiary industries, as well as maintaining human wellbeing. It can be reflected by the minimum household and industrial [water] shortage" (lines 190-2). The minimum household and industrial water shortage cannot reflect the dynamics of industrial revenues and human wellbeing. Depending on the time frame, a system can maximize its economic gain regardless of water shortage. Another example of questionable assumptions appears in line 242: "the production of livestock is also in proportion to its water usage." This is simply wrong unless a Leontief function is used to represent this linear relationship, which is not. Finally, equation 2c aims to minimize a constant, which does not make sense!

Sixth, there are assumptions that limit the generalizability of the model. For example, "In the last stage, the continuous increase of the overload index stimulates the policymakers to alleviate the growth rate of population and GDP" (lines 513-4). Local or state governments in countries that do not follow a central decision-making regime cannot control economic and population growth as a function of ecological health. Even in such regimes, a significant effect of ecological health on economic and population growth is still a strong assumption.

Other, less critical issues: 1. The paper is unnecessarily long. There are many repetitions in the text (e.g. the sentence "food system is the indispensable component for

human lives" has repeated many times). The first three paragraphs of the introduction section explain how important sustainability is. This could reduce to one or two sentences! Similarly, the first three paragraphs of the discussion section talk about the importance of sensitivity analysis. They can reduce to two sentences. 2. It is claimed that Fig. 1 represents a framework of sustainable development. The diagram in Fig. 1 simply shows some causal relationships between modeled variables. How can it represent a sustainable development framework? 3. In line 294, it is claimed that GDP and population always increase, which is simply wrong. 4. "Rational" in Table 1 should be "Reasonable." 5. "Lookup function solved by optimal model" in Table 2 do not add any valuable information. 6. Avoid the use of hyperlinks in the text.

Examples of poor English writing: 1. line 71: "combing" 2. line 204: "alternating" 3. line 208: "improve air pollutions" 4. line 246: "&" 5. line 310: "where OI, PI, CI is" 6. line 465: "finally" (appears twice in a paragraph) 7. lines 469-470: "the economic growth will increase sharply to ensure the local economic development." (circular logic) 8. line 477: "It is easy to understand because . . " (what is easy to understand?) 9. lines 504-5: "the feedback linkage will take effect as that the growing rate of water supply of household and industry (Fig.9d) will miss the rate of water demand" 10. line 511: "the rest water" 11. lines 633-4: "decision-makers are usually hard to choose which one is the better than another one." 12. line 650: "contribute" (should be "attribute"?) 13. line 654: "The parameter $\theta$ and $\alpha$ equal 0.25" 14. line 689: "These individual three items are also prominent aspects or disciplines that contain numerous basic principles." (hard to understand) 15. line 709: "adequately" (should be "adequate") 16. line 712-3: "We need to take advantage of the positive effect as much as possible of a certain model that is, although, usually double-sided." (hard to understand)

---

## Author Comment (AC1) · 7 Jan 2021

Dear referee #1: Thank you very much for giving a decent suggestion of this paper and the opportunity of revising this paper. The suggestions are very useful for me to improve the quality of the paper, and I also quite agree with your comments, in other words, the relationship between SD and optimal model should be clarified. Here are some comments:

1. The concern of SD and optimal model

The overall framework is how to address and quantify the interaction and coevolution of the complex water resources system under the changing external drivers. Therefore, system dynamic model is used to reflect its dynamic status. However, to achieve sustainable development goals, systematic optimization is an indispensable approach because the water usage for socioeconomic development, ecological protection, and food safety conflicts with each other. The two approaches in the current version are indeed not so clear and, so in the revised version, we have to make it clear. Here is the new framework of coupling these two models and will consider it in the revised version of the paper:

As noted in the text, the external drivers of a water resources system can be outlined by the "pendulum model" that can substantially be regarded as the connection of different time steps. Precisely, the "time" variable is inherently the component of SD model and, therefore, it can reveal the dynamic status of a certain system. The external drivers, based on the pendulum model, can be quantified by both the increasing rate of population & GDP size and ecological awareness. They are in conflict with each other and we can regard it as the "pendulum swings" vividly. In the scope of water resources, it can be reflected by the changing water demands. That is, the changing rate of population and GDP size can be revealed by the changing rate of domestic and industrial water demands. The changing water demands will result in the changing water supplies, further changing carrying capacity, food production, etc. It is clear that the external changing condition will inevitably influence the operating state of a system, which can be quantified by SD model that can reveal such relationships and their dependent changes of those variables. Additionally, according to CAS theory, the external changes can also stimulate both the entire system and each agent to readjust themselves and attain the adaptive status, which can be quantified by systematic optimization. However, there's no optimal function within SD model and therefore, the system dynamics of a water system are not optimal and cannot ensure the sustainable development goal. Therefore, it should be coupled with the optimal model.

To couple both models, the initial scheme of water supply should be generated by SD model. We define each time step of external drivers as $\tau$. For each $\tau$, both water supply and demand can be calculated by SD model (For calculation of SD model, the equation of each variable can be seen in Supplementary data of the paper). The water supply scheme generated by SD is the initial solution of the optimal model (Li et al., 2018). The optimal solution (optimal water supply scheme) is generated by the iteration of the optimal model until the adjacent iteration result is less than a specific error. Then, the optimized water supply will transfer back to SD to update the system status of the current time step, and prepare for the next $\tau$ and repeat the whole process. We define T as the total number of $\tau$. If $\tau$<T, repeat the whole process; If $\tau$=T, end the process. The flowchart of the whole process is in the supplement of this reply. (See Fig.1)

2. Calibration and validation of the model

The calibration and validation process is indeed needed as the reviewer mentioned, and we will surely add this part. We have also discussed with each other about this issue (what the result of 2016-2020 represents), and we decided to use these five years for model calibration and validation. The status of 2016-2020 already happens and it is of no significance to simulate this period. 2021~2030 and 2031-2040 can act as different future scenarios (with the different increasing rates of socio-economy) and SD is inherently used to simulate future scenarios. In the revised paper, we will delete the simulation result of 2016-2020, and give new scenarios starting with 2021.

3. Other concerns:

L26: In what way does the paper "highlight the importance of water resources management"? Please explain.

The current water resources system considers multiple water use agents that is conflict with each other. Therefore, in this study, we have generated multiple scenarios based on different weighting factors. Water allocation is not for only one water department but for multiple water use departments, which is an important issue for water resources

allocation for multiple water users. Giving different water resources allocation schemes for different policy-makers from different departments can help them choose the best scheme from portfolios based on their own benefits.

L85: are usually manifested.

Writing mistake. I will make changes in the revised paper.

L113: The framework is not yet raised to theoretical altitude. It is a little bit exaggerating. The word, such as "overall", "general" may be better.

I will make changes in the revised paper. It indeed does not raise to theoretical altitude.

L139-147: Why single out "CAS theory" here? Please explain the usage in this text and the relationship of subsequent sections (or add some relative statements).

CAS theory reveals the system and its components, as well as their relationship. CAS is addressed based on the fact that a system is easy to be influenced by external drivers. There's no doubt that external dynamic changes will influence the status of a system and stimulate the system to make corresponding responses. The response is to make each component and the entire system attain the most suitable status. In the water resources system, the components are multiple water users, i.e., different agents in the scope of systematics. The optimal process is the most effective approach to allocate the water for different water users, making all the agents to their best status, which is the direct manifestation of CAS theory.

L150-158: As noted before, SD and optimal model should be reconsidered. Therefore, this part should be thoroughly edited to reveal the correct logical relationship.

Yes, this relationship should be thoroughly rewritten. The relationship and logic have outlined in my first response to your comments.

L656-657: Please explain why uncertainty analysis "gives a strong reference for decision-making process".

The uncertainty analysis contains scenario analysis based on multiple tradeoffs of water users. Each scenario contains one or two of each water user that share a relatively larger proportion (e.g., A1∼A3 focuses more on ecological streamflow, A4∼A7 focuses more on food safety). By analyzing each water allocation scheme of each scenario, the results can give a strong reference for stakeholders on (1) how to make changes based on the policymakers. If the local policy inclines to ecological protection, then A1∼A3 will be the possible water allocation scheme; (2) how can each stakeholder itself make tradeoffs based on different scenarios. For example, the results show that the equal consideration of each water user can best attain sustainable development, and each stakeholder will make tradeoffs to get a win-win status. It also helps policymakers on how other stakeholder acts if one of the stakeholders is emphasized (with higher weighting factor) of neglected (with lower weighting factor).

L720-721: The general brief sentence should be added to explain how the research framework "becomes a practical tool". The following two paragraphs are used to disclose the general brief sentence. Also, check if the abstract has mentioned this. The title includes "practical tool" and all the text should be surrounded this issue.

Yes, this comment is very important to improve the quality of the paper and help make the logic easy to understand. Actually, the research "becomes a practical tool" can be reflected as I mentioned in the previous reply (L656-657). I will compress those expressions in the corresponding location of the current paper.

Also, I found the overall terminology of the entire framework is a little questionable. The framework of "Economy-ecology-food" (EEF) in which "economy" has already contained "agricultural economy" that corresponds to the "food" module. It has a repetitive concept. I want to rename the framework of "Socioeconomy-ecology-food" (SEF) because the prefix "socio" emphasizes more on "human", and therefore, "socioeconomy" will emphasize more on population, industry, and tertiary industry compared with primary industry.

References:

Li Z, Li C, Wang X, et al. A hybrid system dynamics and optimization approach for supporting sustainable water resources planning in Zhengzhou City, China[J]. Journal of Hydrology, 2018, 556: 50-60.

Please also note the supplement to this comment:
https://hess.copernicus.org/preprints/hess-2020-461/hess-2020-461-AC1-supplement.pdf
* * *
```
        ┌────────────────┐
        │    SD model    │
        └────────────────┘
                │
      ┌───────────────────────┐
      │ External drivers      │
      │ (pendulum model)      │
      └───────────────────────┘
              τ=1
      ┌───────────────────────┐          ┌────────────────┐
      │ Changes in population │          │  Optimal model │
      │ and GDP growth        │          └────────────────┘
      └───────────────────────┘                  │
      ┌───────────────────────┐          ┌───────────────────────┐
      │ Changes in water      │          │ Objectives and        │
      │ demand                │          │ constraints           │
      └───────────────────────┘          └───────────────────────┘
      ┌───────────────────────┐          ┌───────────────────────┐
      │ Influence the feedback│          │ Solution of the model:│
      │ loop                  │          │ decomposition-        │
      └───────────────────────┘          │ coordination          │
      ┌───────────────────────┐          └───────────────────────┘
      │ Influence the water   │ Initial solution
      │ allocation scheme     │────────────┐
      └───────────────────────┘    ┌───────────────────────┐
                                   │ Systematic optimal    │
                                   │ process               │
         ┌────────────┐            └───────────────────────┘
         │ Set output │  N   ◇ Is adjacent iteration ◇
         │ result as  │◄─────  less than specific error?
         │ new initial│            │
         │ solution   │            Y
         └────────────┘   ┌───────────────────────┐
                          │ The optimal allocation│
                          │ scheme in current     │
                          │ time step τ           │
      ┌───────────────────────┐
      │ Update the system     │
      │ status                │
      └───────────────────────┘
              ◇ τ=T? ◇  ── Y ──► ┌────────┐
                                  │  End   │
                N                 └────────┘
      ┌───────────────────────┐
      │ Encounter the next τ  │
      │ (Here τ=τ+1)          │
      └───────────────────────┘
```

**Fig. 1.**

---

## Author Comment (AC2) · 8 Jan 2021

Dear referee #2:

Thanks for your decent comments on my paper and your comments are very helpful to improve the quality of the paper. Here is my point-to-point reply to your comments:

1. First, it is not clear what the contribution of the paper is. Is the paper trying to introduce a new method? Is it trying to solve a problem? Is it trying to introduce a new theory? The paper needs to have a clear goal and then focus on that. If the goal is to

introduce a new method, it should provide clear information on how optimization and system dynamics can be integrated. The current manuscript does tell us much about the integration process, as explained elegantly by the first reviewer. If the goal is to solve a problem, the problem should be defined. The current presentation provides some predictions about some measures and explains their dynamics, but it is unclear what problem it is trying to solve. Is the problem unsustainability? Is it depletion of resources? What are the research questions? What is the hypothesis that the paper aims to test? And then, it must provide some robust solutions. The paper has some scattered solutions in the discussion section, but they need to be organized and become the focus of the paper if the goal is to solve a problem. Finally, if the goal is to advance a theory, the whole writing should support that goal. The current manuscript aims to achieve all of these goals, and that makes it ambiguous.

Reply: Yes, this is the root of the motivation of why we propose this study and it is of most importance. A research paper should define a research question and we have to develop a certain approach to solve this question. Well, the existed problem of water resources is the unsustainability uses, reflected by quick consumption of water resources especially on socio-economy, neglecting ecological streamflow water. However, as a research paper, a common problem should be compacted into a research problem. Based on the literature review, systematic approach and nexus thinking has developed to solve the problem considering multiple uses of water resources, some of which using an advanced approach, such as systematic optimal approach. However, few studies put emphasize on optimal water allocation in a dynamic way (Optimal approach is one of the effective ways to attain sustainable water uses, but usually give water allocation scheme in a static way, i.e., a certain future year), or, system dynamics on simulating future dynamics on water resources system rarely considering sustainable (or optimal) water uses (Li et al., 2019; Yang et al., 2019). This is the research problem. Therefore, based on this, we develop the current study and introducing this integrating method. As the first reviewer mentioned, such a framework cannot be considered as a theory, it is exaggerating, and we will delete "theory" work in the text.

[Figure]

2. Second, the paper is hard to follow mainly because its English writing is poor. There are many instances of grammatical and typing errors, as well as bad writing. I have listed some of the examples at the end of this note.

Sorry for our poor English writing. We are from the country that the mother tongue is not English. But we will thoroughly modify the English grammar and expressions and make it better, including but not limited to revise it by some person whose mother tongue is English. Also, I will also consider the listed examples.

3. Third, the paper does not get the system dynamics terminology right. The method is mistakenly called "system dynamic," which is wrong. Another instance is the use of "systematics" as a misnomer of "systems perspective" or "systems relationships." Yet another example is the use of "cause-and-effect feedback loop" (Fig. 6 caption) instead of "causal loop diagram." Also, some statements in the manuscript make me nervous and skeptical about the authors' command of system dynamics modeling. They claim: "if the water supply increases at the same rate as water demand caused by increased socio-economic index, this feedback will be the positive feedback regulation that results in the polarization because the ecological water will be occupied and environmental protection will not be guaranteed." Note the use of "if" and "this feedback will be" in this statement. A feedback loop cannot be conditional; it is either positive or negative. The ambiguity of a feedback loop indicates a major deficiency in the model.

Reply: Some ambiguous expressions will be deleted in the revised manuscript. I will also unify all the terms as "SD model". System dynamic model includes many variables and their mutual relationships. The "time" variable is the inherent variable that links to other variables, that's why SD model can run dynamically. Feedback loop is the inherent function of SD model, but maybe some expression (such as "if the water supply increases at the same rate as water demand caused by increased socio-economic index, this feedback will be the positive feedback regulation that results in the polarization because the ecological water will be occupied and environmental protection will not be guaranteed.") is ambiguous, maybe I did not express the correct information in

the text. This part must be thoroughly rewritten.

As SD model is used to simulate the dynamic operating status, the writing will put more emphasis on how it simulates the dynamic system and how it coupled with the optimal model. In Section 2 I will outline how to couple SD and optimal model (See my reply to both RC1 and the next point). In the next section I will rewrite this SD part, includes the variable types, how to connect the "time" variable, and relationships, to reveal how SD can simulate the system over time. Actually, Fig5 can represent the system dynamics diagrammatically, which can be revealed by SD. I will add the following statements to the paper:

"The essence of system dynamics is first-order differential equations, which is mainly composed of four basic variables: Level variables, Rate variables, Auxiliary variables, and flows. The level variable describes the cumulative effect of the system and reflects the accumulated amount that changes over time. It can be regarded as the storage of information and is generally represented by a rectangular box. The value of the state variable is the sum of the net inflow or outflow rate in the previous time and the corresponding simulation step size. The rate variable reflects the speed of the cumulative effect of the system and the change of the state variable over time, so it represents the speed of the system change or the amplitude of the decision. It embodies the control of the state variable and can determine the input and output variables in the state variable. Auxiliary variables are the intermediate variables throughout the decision-making process, which can be transformed from input variables to output variables through mathematical expressions. Flow refers to the relationship between variables, and it's the bridge connecting any variable, either material flow or information flow, which reflects the behavior between the system or variables. All these variables are connected with certain equations/functions, and therefore all the variables compose a large system. As level variables is linked over time, and "time" variable is usually linked with other variables, SD will be able to operate a large system in a dynamic way over time."

4. Fourth, the presented model is very confusing. It is not clear if the model is trying

to "characterize" the real system or represent an ideal situation. If the goal is to characterize the real system, then a system dynamics model would be sufficient. Agents of a real system rarely optimize things. If the goal is to represent an ideal situation (in a neoclassical sense), then the optimization model would be sufficient. I believe this confusion arises because the goal of the modeling is not defined adequately. As the first reviewer explained, the relationships between the optimization and the SD model need to be clarified.

Reply: Well, SD can simulate a system in a dynamical way but it has no optimal function, that's why we couple SD and optimal model. The referee states "I believe this confusion arises because the goal of the modeling is not defined adequately", yes, in the introduction I exactly not define the research question clearly, and I replied to this issue in point 1. That is, to achieve the sustainable development goal (because water resources management in some of the region is still unsustainable water uses), an optimal approach is an indispensable tool to deal with the problem. However, as accelerated consumption of water resources is happening and, more water users should be considered, advanced requirements have been put forward, that is, more systematic approaches should be adopted. (Address research question) Current studies most manage water resources in a static way, instead of dynamic, which cannot reach the new advanced requirement of water resources management (Li et al., 2019). So, we have to couple both models in this study.

To couple both models, the initial scheme of water supply should be generated by SD model. We define each time step of external drivers as $\tau$. For each $\tau$, both water supply and demand can be calculated by SD model (For calculation of SD model, the equation of each variable can be seen in Supplementary data of the paper). The water supply scheme generated by SD is the initial solution of the optimal model (Li et al., 2018). The optimal solution (optimal water supply scheme) is generated by the iteration of the optimal model until the adjacent iteration result is less than a specific error. Then, the optimized water supply will transfer back to SD to update the system status of the

current time step, and prepare for the next $\tau$ and repeat the whole process. We define T as the total number of $\tau$. If $\tau<T$, repeat the whole process; If $\tau=T$, end the process. The flowchart of the whole process is in the supplement of this reply.

5. Fifth, the presented model has some questionable assumptions. For example, it assumes that "The goal of the economic agent is aiming at increasing revenue of secondary and tertiary industries, as well as maintaining human wellbeing. It can be reflected by the minimum household and industrial [water] shortage" (lines 190-2). The minimum household and industrial water shortage cannot reflect the dynamics of industrial revenues and human wellbeing. Depending on the time frame, a system can maximize its economic gain regardless of water shortage. Another example of questionable assumptions appears in line 242: "the production of livestock is also in proportion to its water usage." This is simply wrong unless a Leontief function is used to represent this linear relationship, which is not. Finally, equation 2c aims to minimize a constant, which does not make sense!

Reply: (1) At the beginning, I wrote this sentence "The goal of the economic agent is aiming at increasing revenue of secondary and tertiary industries, as well as maintaining human wellbeing. It can be reflected by the minimum household and industrial [water] shortage" just for enrich the paper. It does seem to address the wrong assumption at this time. Now it is the redundant expression. I will delete this sentence. (2) The reason why I added "meat production" is that I thought that people not only eat vegetables but also meat. But now, vegetables and crops account for relatively more proportion. Meat production not only influenced by water uses but also nutrients (it is out of the scope of water resources). I will delete the "meat production" part. (3) AAPFD is not constant. It is a function of observed flow and actual ecological streamflow. I addressed in the supplementary data. For clarity, I will rewrite the whole expression in equation 2c.

6. Sixth, there are assumptions that limit the generalizability of the model. For example, "In the last stage, the continuous increase of the overload index stimulates the

policymakers to alleviate the growth rate of population and GDP" (lines 513-4). Local or state governments in countries that do not follow a central decision-making regime cannot control economic and population growth as a function of ecological health. Even in such regimes, a significant effect of ecological health on economic and population growth is still a strong assumption.

Reply: the future economic growth rate is, actually, the different development scenarios. For dynamic simulations in the future, scenario analysis is definitely a powerful and effective way (Haasnoot et al., 2011; Yang et al., 2019). So, we generated 3 stages as different scenarios to simulate the dynamics based on different economic growth rate. Maybe the statement "In the last stage, the continuous increase of the overload index stimulates the policymakers to alleviate the growth rate of population and GDP" is a little bit questionable, but, we just want to convey the information that "lower growth rate of the economy can alleviate the overload index of an entire water resources system".

Critical issues:

1. The paper is unnecessarily long. There are many repetitions in the text (e.g. the sentence "food system is the indispensable component for human lives" has repeated many times). The first three paragraphs of the introduction section explain how important sustainability is. This could reduce to one or two sentences! Similarly, the first three paragraphs of the discussion section talk about the importance of sensitivity analysis. They can reduce to two sentences.

Reply: I will make changes to shorten the paper

2. It is claimed that Fig. 1 represents a framework of sustainable development. The diagram in Fig. 1 simply shows some causal relationships between modeled variables. How can it represent a sustainable development framework?

Reply: Fig.1 is used to address the detail of the system (agents). Maybe the name of the picture is not suitable. I will delete this figure and list a table to imply the content of

the integrated system.

3. In line 294, it is claimed that GDP and population always increase, which is simply wrong.

Reply: GDP and population not "always" increase, it will increase for long-term time periods (and will stay still or decrease) and the simulation time is only for 25 years, not forever. The pendulum dynamics also do not hold for long time scales, as the editor notes. I will delete the "always" word.

4. "Rational" in Table 1 should be "Reasonable."

Reply: I will make changes.

5. "Lookup function solved by optimal model" in Table 2 do not add any valuable information.

Reply: I will delete this issue. This question arises because the relation of SD and optimal model is unclear in the current paper. The new framework of coupling SD and optimal model have nothing to do with the "Lookup function solved by optimal model" in the current version. How to couple these two models is addressed in the reply of RC1 and point 4. Also, see the figures at the end of this reply.

6. 6. Avoid the use of hyperlinks in the text.

Reply: I will make changes.

References:

Haasnoot M, Middelkoop H, Van Beek E, et al. A method to develop sustainable water management strategies for an uncertain future[J]. Sustainable Development, 2011, 19(6): 369-381.

Li T, Yang S, Tan M. Simulation and optimization of water supply and demand balance in Shenzhen: A system dynamics approach[J]. Journal of Cleaner Production, 2019,

207: 882-893.

Yang Z, Song J, Cheng D, et al. Comprehensive evaluation and scenario simulation for the water resources carrying capacity in Xi'an city, China[J]. Journal of environmental management, 2019, 230: 221-233.

Please also note the supplement to this comment:
https://hess.copernicus.org/preprints/hess-2020-461/hess-2020-461-AC2-supplement.pdf

**Fig. 1.**

```
        ┌─────────────┐
        │  SD model   │
        └──────┬──────┘
               ↓
      ┌──────────────────┐
      │ External drivers │
      │ (pendulum model) │
      └────────┬─────────┘
             τ=1                              ┌──────────────┐
      ┌──────────────────┐                    │ Optimal model│
      │ Changes in       │                    └──────┬───────┘
      │ population and   │                           ↓
      │ GDP growth       │                    ┌──────────────┐
      └────────┬─────────┘                    │ Objectives   │
      ┌──────────────────┐                    │ and          │
      │ Changes in water │                    │ constraints  │
      │ demand           │                    └──────┬───────┘
      └────────┬─────────┘                    ┌──────────────────────┐
      ┌──────────────────┐                    │ Solution of the      │
      │ Influence the    │                    │ model: decomposition-│
      │ feedback loop    │                    │ coordination         │
      └────────┬─────────┘                    └──────┬───────────────┘
      ┌──────────────────┐  Initial solution         ↓
      │ Influence the    │───────────────────►┌──────────────┐
      │ water allocation │                    │ Systematic   │
      │ scheme           │                    │ optimal      │
      └────────┬─────────┘                    │ process      │
                        ┌──────────────┐      └──────┬───────┘
                        │ Set output   │  N          ↓
                        │ result as new│◄─── Is adjacent iteration
                        │ initial sol. │     less than specific error?
                        └──────────────┘           │ Y
                                          ┌──────────────────────┐
      ┌──────────────────┐                │ The optimal          │
      │ Update the system│◄───────────────│ allocation scheme in │
      │ status           │                │ current time step τ  │
      └────────┬─────────┘                └──────────────────────┘
               ↓
           τ =T? ───Y───► End
               │ N
      ┌──────────────────┐
      │ Encounter the    │
      │ next τ(Here      │
      │ τ=τ+1)           │
      └──────────────────┘
```

**Supplement:**